# Prediction-error signals in anterior cingulate cortex drive task-switching

Nicholas Cole[1], Matthew Harvey[1], Dylan Myers-Joseph[1], Aditya Gilra [2,3] &
Adil G. Khan [1] ✉

Task-switching is a fundamental cognitive ability that allows animals to update their knowledge of current rules or contexts. Detecting discrepancies between predicted and observed events is essential for this process. However, little is known about how the brain computes cognitive prediction-errors and whether neural prediction-error signals are causally related to task-switching behaviours. Here we trained mice to use a prediction-error to switch, in a single trial, between responding to the same stimuli using two distinct rules. Optogenetic silencing and un-silencing, together with widefield and two-photon calcium imaging revealed that the anterior cingulate cortex (ACC) was specifically required for this rapid task-switching, but only when it exhibited neural prediction-error signals. These prediction-error signals were projection-target dependent and were larger preceding successful behavioural transitions. An all-optical approach revealed a disinhibitory interneuron circuit required for successful prediction-error computation. These results reveal a circuit mechanism for computing prediction-errors and transitioning between distinct cognitive states.

Animals need to rapidly update their behaviour to survive in a changing environment, and such behavioural flexibility is studied experimentally using task-switching paradigms. Task-switching involves shifting between distinct cognitive rules or contexts, allowing flexible behaviour adapted to changing environmental demands[1]. The framework of predictive processing[2–4] provides a simple yet powerful way of describing flexible task-switching behaviour using three stages. First, animals maintain a previously learned prediction of what they expect to happen in the world at any given instant, i.e., a 'model of the world'. Second, they may detect discrepancies between predicted and observed events, a 'prediction error'. Third, the prediction error guides the updating of their model of the world and their ongoing behaviour. While this account has widespread support across humans, monkeys and rodents[2,5,6], there is no clear link between the neural representations of prediction errors and their causal requirement in updating mental rules and subsequent behaviour.

There is evidence for cognitive prediction-error signalling in the prefrontal cortex (PFC) in tasks across humans, monkeys, and rodents[7–13]. The ACC in particular plays a central role in detecting the need for updating behavioural rules, and implementing the updated rules[7–10,14–25]. However, it is unclear how prediction-error signals are distributed across the cortex, how these signals are organised with respect to the projection targets of cortical neurons, and to what extent the amplitude of prediction-error responses is important in subsequent behaviour. Crucially, it is unclear whether prediction-error signals have any causal influence on the subsequent updating of an animal's behavioural strategy. Finally, the inhibitory circuit basis for computing cognitive prediction errors is largely unknown.

Determining the neural basis of cognitive transitions is challenging because it requires experimental control over an animal's internal model of the world. Animals need to demonstrably hold one rule in mind and transition to another distinct rule on noticing a violation of a prediction based on the current rule. Although animals can be trained on tasks which change or reverse rules in a block-wise manner[12,26–33], these transitions typically involve several tens to hundreds of trials of intermediate performance, making it difficult to directly relate neural

---

[1]Centre for Developmental Neurobiology, King's College London, London, UK. [2]Machine Learning Group, Centrum Wiskunde & Informatica, Amsterdam, the Netherlands. [3]Department of Computer Science, The University of Sheffield, Sheffield, UK. ✉e-mail: khan.adil@kcl.ac.uk

activity to an update of cognitive rules. One-shot block transitions provide a significant advantage in this regard. In one-shot block transitions[34,35], a single error leads to a complete and persistent switch between distinct cognitive states. Thus, repeated one-shot block transitions within a session would allow directly relating neural prediction-error responses with the updating of mental rules.

The identification of temporally well-defined cognitive prediction errors would allow addressing a key question: What neural circuit compares predictions and observations to compute prediction errors? Computing prediction errors requires inhibition[2], and distinct inhibitory cell classes with their specific connectivity patterns provide the basis for current models of prediction error circuits[36]. In particular, vasoactive intestinal peptide (VIP) expressing interneurons are crucial in shaping cortical activity[37–39] and provide unique computational opportunities through a disinhibitory motif involving somatostatin (SOM) expressing interneurons[40]. While VIP interneurons have been hypothesized to be a key player in prediction error computation[36], directly testing this theory is challenging, as it would require measuring neural prediction errors while the activity of VIP interneurons is perturbed in animals performing a cognitive task. Importantly, any neural perturbation should not lead to a change in the task-switching behaviour itself, as this would confound any interpretation of the neural circuit consequences of the perturbation.

Here we studied the neural basis of cognitive prediction errors during task-switching in the mouse neocortex. We trained mice to perform largely one-shot block transitions triggered by cognitive prediction errors. The prediction error was an absence of an expected stimulus, allowing us to better isolate cognitive factors from reward and stimulus evoked factors[41]. Using behavioural modelling, widefield calcium imaging, and optogenetic silencing, we generated a cortex-wide map of prediction-error signals and established a specific role for the ACC in task-switching behaviour. Using two-photon imaging of the ACC we identified a population of prediction-error neurons and, crucially, established that the duration of the prediction-error signal corresponded with the requirement of the ACC in task-switching at timescales within and across trials. We finally used all-optical methods to bi-directionally modulate VIP interneuron activity and established that VIP interneurons play a specific and causal role in the computation of prediction errors in the ACC. These results provide direct evidence for a causal role of prediction-error signals in the ACC in task switching, and identify a key inhibitory interneuron class required for cognitive prediction error computation.

## Results

We trained head-fixed mice to switch between blocks of discriminating two visual stimuli or discriminating two olfactory stimuli (Fig. 1A, B). During olfactory discrimination, a random subset (70%) of odour stimuli were preceded by the same visual stimuli, now irrelevant to the task. Mice repeatedly switched between attending to and accurately discriminating the visual stimuli (rule 1) and ignoring the same visual stimuli while accurately discriminating the odour stimuli (rule 2). Mice performed up to 15 behavioural switches, or block transitions per session (Fig. 1C, range 6 to 15, median 14 transitions across 13 sessions, 10 mice, one session per mouse shown. Number of trials per block, median ± IQR, 33 ± 1 and 35 ± 8 trials in visual and odour blocks respectively). Visual and odour stimuli were never presented simultaneously (Fig. 1B, Supplementary Fig. 1).

### One-shot cognitive task-switching behaviour
Our aim was to repeatedly capture the transition between two distinct and accurately applied task rules within a well-defined time period. Furthermore, to better separate cognitive processes from stimulus- or reward-evoked activity during the transition, we needed the block transitions to be inferred without any explicit stimulus or reward signal. Our task satisfied these requirements on the transitions from

odour to visual blocks: mice noticed the absence of an expected odour stimulus (an odour prediction error) to switch their behaviour, and accurately responded to the now-relevant visual stimuli in subsequent trials (Fig. 1D, E). Thus, we focused on the transition from odour to visual blocks, triggered by the omission of an expected odour stimulus (but see below for visual to odour block transitions).

Animals typically require many trials to switch between blocks of distinct rules, transitioning through periods of intermediate performance[12,26–32,42]. This makes it difficult to precisely identify the behavioural and neural processes underlying a rule-switch. Optimal 'one-shot' transitions offer a significant advantage, since they involve animals switching between two accurately applied rules after a single error[34,35]. Most odour to visual block transitions in our task were indeed one-shot (Fig. 1D–F, 63.2% one-shot transitions, see Supplementary Movie 1 for an example). We used strict criteria to ensure that the one-shot block transitions captured a complete mental transition between two demonstrably applied rules (see methods). A low probability of licking in response to irrelevant visual stimuli in the odour block led to a corresponding proportion of zero-shot block switches, or 'fluke transitions' (Fig. 1F, zero trial bin) which were removed from further analysis. The remaining transitions were dominated by one-shot transitions (Fig. 1F arrowhead) with a smaller fraction requiring two or more error trials.

To understand the processes underlying this rapid task-switching, we first fitted the behaviour to a basic reinforcement learning (RL) model (tabular SARSA, Fig. 1G, top) in which an agent continually updated its estimated value of licking in response to stimuli in each trial (the Q-value). This model was unable to reproduce the rapid block transitions observed in the data (Fig. 1H, top), and the best fit of learning and exploration rates produced slower block transitions (Fig. 1H, middle, see methods). We next built an RL model with a context belief state, hierarchically controlling subsequent choices (Fig. 1G, bottom). In this model the agent updated its belief about which context (or block) it was in by a context error signal. This model learnt two Q-value-tables for the distinct contexts, capturing the two rules of the task. Once the learning was complete, the agent was able to rapidly switch between these two Q-value-tables on accumulating a large enough context-error signal. This belief state model reproduced the rapid block transitions (Fig. 1H, bottom). Once the model had learnt the task, freezing all learning, i.e., Q-value updates, did not affect the block transitions (see methods), demonstrating that switching between blocks did not involve any further learning. Instead, the agent inferred the block transitions using the context error. These results suggest that rapid task-switching behaviour involves transitioning between abstract representations of rules or contexts, driven by prediction errors.

### Map of prediction error signals
Which brain regions signal the prediction error at block transitions? We measured neural activity across the entire dorsal cortical surface using widefield calcium imaging (Fig. 2A) to identify the cortical regions which represent cognitive prediction-error signals at block transitions. Our task design provides a well-defined moment of prediction error at the odour to visual block transition: when, following the visual stimulus, an expected odour does not arrive (Fig. 1D). Any neural activity specific to this prediction error would be (1) absent in the final trials of the odour block when the odour was expected and actually delivered, (2) present when the odour was expected but absent (the prediction error), and (3) absent once again when the odour was no longer expected later in the visual block (schematic of the three conditions shown in Fig. 2B).

We aligned the hemodynamic-corrected and movement-corrected calcium activity to these three temporal epochs of odour onset, odour prediction error, and no odour predicted (Fig. 2B). The activity in these epochs evolved across the cortex over time, with

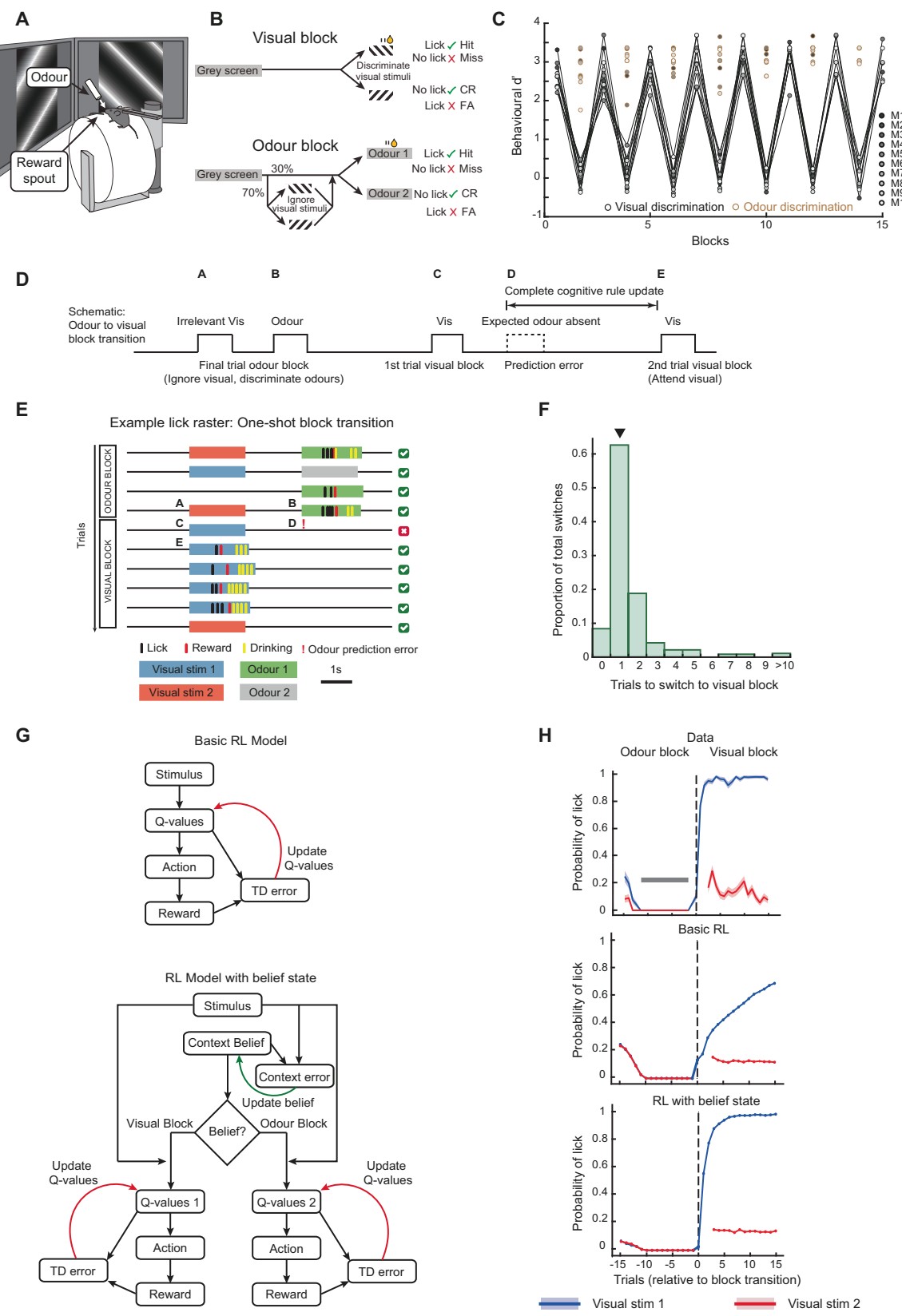

higher activity apparent in the odour prediction-error condition (Fig. 2C, first three columns). We identified pixels at each time point which satisfied the criteria described above (see methods) and mapped their activity (Fig. 2C final column, colour map represents t-statistic of comparison between first and second column). This revealed the precise spatio-temporal evolution of the prediction-error

signal across dorsal cortex, which spread from anterior secondary motor regions to posterior parietal areas (average map from 1.2 to 2.7 s shown in Fig. 2D). In particular, we found a strong prediction-error signal in areas superior to prefrontal cortex. Although the visual cortex and olfactory bulb showed robust sensory stimulus evoked responses to visual and odour stimuli respectively, they did not contain

**Fig. 1 | Mice perform rapid task-switching with one-shot block transitions.**
**A** Experimental setup. **B** Task schematic. Mice switched between blocks of discriminating two visual stimuli in the visual blocks or discriminating two olfactory stimuli while ignoring the same visual stimuli in the odour blocks. **C** Behavioural discrimination performance (behavioural d') across blocks (N = 10 mice). Shades of grey indicate individual mice. Odour discrimination performance is shown in brown circles. Mice performed up to 15 block transitions in a session. **D** Schematic of a transition from an odour block to a visual block, showing the last trial of the odour block and first two trials of the visual block, indicating the moment of odour prediction error. Labels A-E refer to timepoints in **E**. During one-shot block transitions as in **E**, a complete cognitive rule update occurs in the time indicated. **E** Example behaviour from an odour to visual block transition (lick raster) showing stimuli, lick, and reward times. In the odour block the mouse ignored both visual gratings while accurately discriminating odour stimuli, but switched rules after a single error trial and started accurately discriminating the same visual stimuli. **F** Histogram showing

the number of trials required to switch between the two blocks, with one-shot transitions indicated by an arrowhead. N = 95 transitions, 17 sessions, 14 mice. **G** Schematic of reinforcement learning (RL) models. Top, basic RL, bottom, RL with belief state. In the RL model with belief state, Q-values were only updated while the agent learnt the task, following which only context belief was updated to switch between blocks. **H** Average probability of licking the reward spout in response to Visual Stimulus 1 (rewarded only in visual blocks) and Visual Stimulus 2 (unrewarded in both blocks), aligned to the block transitions. Top, data from average of 95 odour to visual block transitions. Shading indicates SEM. Grey bar indicates the 10-trial duration where mice performed at 100% accuracy with no licks to either irrelevant visual stimulus, which was a condition for triggering a block transition. First 3 trials of each block were forced to be Visual Stimulus 1 (to assess the animal's belief of which block it was in), resulting in a gap in the red curve depicting Visual Stimulus 2. Middle, basic RL model fit to data. Bottom, RL model with belief state fit to data. Source data are provided as a Source Data file, for this and subsequent figures.

detectable prediction-error signals (Fig. 2D, averaged activity PSTHs shown in Fig. 2E). Together these results suggested that prefrontal cortical areas, not primary sensory areas, are involved in detecting the cognitive prediction error relevant for the rapid block transitions in this task.

## ACC is required for rapid block transitions

We next asked which specific region within prefrontal cortex was required for driving the task-switching behaviour. A number of studies have linked ACC activity to surprising events[9,42], errors[10,23] or negative feedback[43,44], which promote updating of behavioural policies[16,21], or promote control[15]. In addition, the prelimbic (PL) cortex has been implicated in task-switching behaviour[45]. If prediction-error signalling in the ACC or PL is required for rapid task-switching, silencing these regions should disrupt the behaviour specifically at block transitions. To test this, we optogenetically silenced the ACC or PL bilaterally for the entire behavioural session in a group of mice (Fig. 3A). ACC silencing caused a strong deficit in switching from odour to visual blocks, when the mice needed to detect the absence of an expected odour to switch rules. Strikingly, with the ACC silenced, mice ignored the rewarded visual stimulus repeatedly in anticipation of an odour stimulus, reflecting the continuing application of the odour block rule (Fig. 3B). As a result, ACC silencing increased the number of trials taken to switch (Fig. 3C, D) and reduced the proportion of one-shot odour to visual block transitions (Fig. 3D, E). This was also the case when silencing the ACC using bilateral infusions of the GABA-A receptor agonist muscimol (Supplementary Fig. 2C). The belief-state RL model could fit the ACC silencing data with the prediction-error signal reduced by a factor of 0.22, suggesting that the task-switching deficit could be explained by a partial reduction in the ACC prediction-error signal (Supplementary Fig. 2D). These results demonstrate that the ACC is required for processing the omission of an expected event, a prediction error, during task-switching.

Interestingly, silencing PL in the same mice did not affect switching behaviour (Fig. 3C). Although we observed a cognitive prediction-error signal across a substantial portion of frontal cortex in our widefield experiment (Fig. 2D), the absence of any behavioural effect of silencing PL, adjacent to the ACC, reveals a striking specificity in the role of the ACC. These results indicate that the ACC has a specific role in rapidly updating behavioural rules or contexts within seconds, and this role is not widely shared with other prefrontal areas.

While the ACC is required for rapidly switching between task rules, does its role extend to applying these task rules once the animal has switched blocks? We found that even during continuous ACC silencing, once the mice did spontaneously switch from ignoring to discriminating the visual stimuli, the subsequent accuracy of visual discrimination was only slightly lower from unsilenced sessions

(Fig. 3F). This suggests that the ACC plays a critical role in transitioning between task rules, and less so in maintaining these rules.

In addition to rapid task-switching, the ACC has been implicated in guiding slower learning processes[46–48]. We optogenetically silenced the ACC in a subset of mice as they first learned the switching task and found no difference relative to controls in either the rate of learning a novel stimulus-reward association, or learning to ignore irrelevant stimuli[49] (Supplementary Fig. 2G). Thus, in this paradigm, the role of the ACC is specific to rapid task-switching and does not extend to slower learning.

Overall, these results established that the ACC is essential for rapid task switching driven by a cognitive prediction error. However, ACC projections become dispensable when task demands are reduced in an attentional task[50]. We asked if a similar result was true in task-switching behaviours, that is, if an animal could overcome the inhibition of the ACC if the block transition was marked by a more salient event than the absent expected odour. The opposite direction of block transition in our task, from visual to odour blocks, was marked by the unexpected arrival of an odour, a more salient prediction error. Most visual to odour block transitions (60%) were also one-shot (Supplementary Fig. 3A, B) and were captured well by the belief-state RL model (Supplementary Fig. 3C). Interestingly, silencing the ACC had no effect on the switching behaviour in these visual to odour block transitions (Supplementary Fig. 3D–F). Thus, when task demands are reduced by using a highly salient event to mark block transitions, the ACC may become dispensable in task switching, similar to attentional tasks. Our study, however, focused on the odour to visual block transitions, which relied on the ACC.

## Prediction-error signals in ACC neurons

Since block transitions were typically complete within a single trial, this also meant that the neural activity in the ACC required for the block transitions must be present within the few seconds between the prediction error (absent predicted odour) and the next visual stimulus (Fig. 1D). We therefore asked what neural signals in the ACC during this period may account for its role in rapid block transitions. We recorded the activity of populations of ACC neurons during the behaviour using chronic two-photon calcium imaging through a microprism (Fig. 4A, Supplementary Fig. 4A). As expected, we found that individual ACC neurons responded to many task-related variables. Subsets of neurons responded to visual and odour stimuli, locomotion onsets, reward delivery and licking (Fig. 4B, C). A binary classifier was able to accurately decode the identity of visual and odour stimuli after stimulus onsets from ACC neural population activity, and was further able to decode the block type both before and after stimulus onsets (Fig. 4D). Thus, ACC contained diverse signals relevant to the ongoing task.

To identify neurons representing prediction errors, we asked if a neuron showed responses which were (1) absent in the final trials of the odour block (2) present during the odour prediction error, and (3)

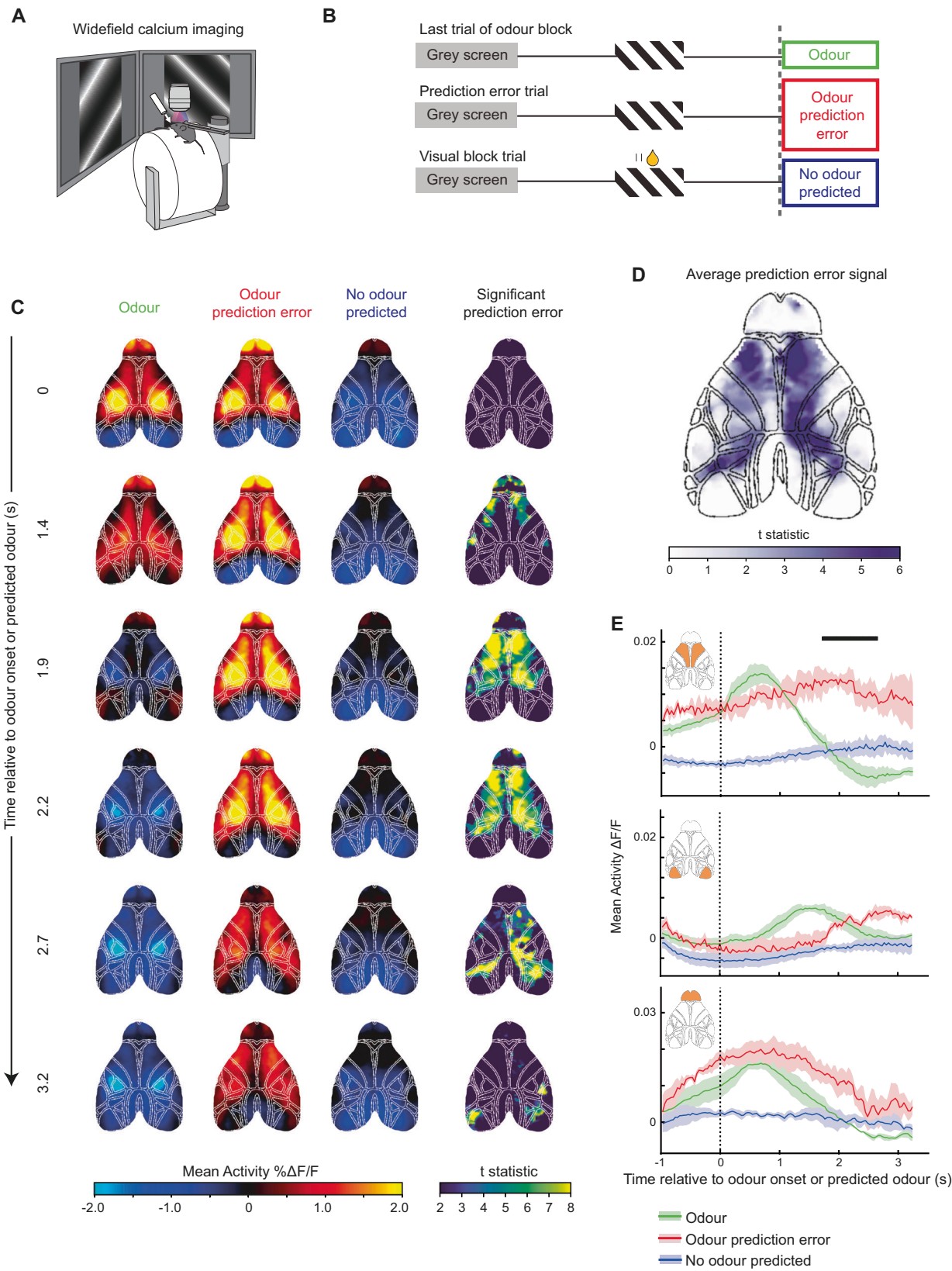

absent once again when the odour was no longer expected later in the visual block (Fig. 4E, the same criteria as used in the widefield data). We found neurons in the ACC that satisfied these criteria, which we termed 'prediction-error neurons'. Figure 4F shows an example prediction-error neuron which was suppressed in trials when the expected odour did arrive, but responded strongly during the odour prediction error

(when an odour was expected but did not arrive), and was not responsive later when an odour was no longer expected (see also Supplementary Fig. 4B). Importantly, the prediction-error signal is a response to the non-occurrence of an expected stimulus, not a stimulus onset or offset, ensuring that the response originates from a violated stimulus expectation.

**Fig. 2 | Cortex-wide map of prediction-error signalling. A** Schematic of widefield calcium imaging. **B** Schematic of behaviour during a one-shot transition from an odour to visual block, indicating the three trial types used to identify prediction-error pixels. The first trial of the visual block contains an odour prediction error, due to the absence of an expected odour stimulus, which mice can use to infer the change in block type. Prediction-error pixels were defined as pixels with significantly different activity between the odour prediction-error period and both the odour period and no odour predicted period. **C** Mean activity (columns 1-3) and significant prediction-error pixels (column 4) across 6 timepoints, aligned to expected/actual odour onset ($N = 11$ sessions, 4 mice). Significance column shows

pixels with activity significantly different between odour prediction error and odour trials, and between odour prediction error and no odour predicted trials, with the t-statistic values from the comparison of column 1 and 2 displayed in colour code. **D** Pixels with significant prediction-error signalling, averaged across 1.2–2.7 s relative to the expected/actual odour onset. **E** Mean activity profiles from the secondary motor cortex (top), primary visual cortex (middle) and olfactory bulb (bottom, ROIs shown in insets). Shading indicates SEM. Black line indicates periods during which the odour prediction-error response was significantly different from both the odour and no odour conditions (two-sided paired t-test, $P < 0.01$ for more than 500 ms, $N = 4$ mice).

The average of all prediction-error neurons positively activated by the prediction error is shown in Fig. 4G. The percentage of all recorded neurons which showed any prediction-error responses was 9% (Fig. 4H). This group contained neurons with a positively activated response to the prediction error (Fig. 4F, g, 27% of prediction-error neurons), as well as a similar number of neurons with an inhibited response to the prediction error (30% of prediction-error neurons, Supplementary Fig. 5A) and other combinations of activity profiles. Response inhibition by the arrival of the expected odour occurred for both rewarded and non-rewarded odours, suggesting that these neurons did not reflect a simple negative reward signal (Supplementary Fig. 5B). Crucially, changes in running and licking could not account for the identification of prediction-error neurons (Supplementary Fig. 5C, D).

To investigate whether prediction-error neurons were found widely across the brain, we conducted the same experiment using two-photon calcium imaging to record activity in the primary visual cortex (V1) (Fig. 4I, left). Consistent with the widefield calcium imaging results, our findings revealed a near absence of prediction-error neurons in V1 (Fig. 4I, right). This corroborated the widefield imaging results and, importantly, confirmed that our identification of prediction-error neurons in the ACC was statistically reliable.

We also studied the opposite direction of block transition, from visual to odour blocks, which was marked by the unexpected arrival of an odour stimulus. We again found prediction-error neurons in the ACC which responded differently to the unexpected odour, compared to the same odour when expected, or no odour (Supplementary Fig. 6A, B, similar proportions of prediction-error neurons were obtained when controlling for running and licking behaviours, data not shown). Crucially, however, we obtained a similar proportion of these neurons in V1 (Supplementary Fig. 6C). Thus, prediction-error neurons signalling an unexpected appearance of an odour are found both within and outside prefrontal cortex, consistent with the finding that the ACC is not necessary for this direction of transition (Supplementary Fig. 3D–F). Interestingly, neurons which represented both directions of prediction error were present at a higher proportion than expected by chance (Supplementary Fig. 6D), suggesting the presence of generalist, in addition to specialist prediction-error neurons in the ACC.

**Striatal projections in the ACC exclude prediction errors**

Prediction-error neurons may broadcast their activity widely across the brain, or they may be enriched or excluded from populations based on their projection target. We distinguished between these two scenarios in the same mice by selectively labelling the sub-population of ACC neurons which projected to the striatum, a major projection target of the PFC[51]. CTB-Alexa647 injections in the striatum identified striatal-projection neurons in the ACC (Fig. 4J), and non-retrolabelled neurons were enriched for non-striatal projecting neurons. Although striatal projecting neurons had mostly overlapping response properties with non-striatal projection neurons (Fig. 4K, Supplementary Fig. 5E, F), they contained a significantly smaller proportion of prediction-error neurons (Fig. 4L, Chi-squared test of proportion $P = 0.0015$, striatal projecting prediction-error

neurons = 21/410, 5%, non-striatal projecting prediction-error neurons = 463/4888, 9%). This lower proportion of striatal projecting prediction-error neurons was outside the 99% confidence intervals of the non-striatal projecting prediction-error neurons (Fig. 4M, bootstrap test). Thus, prediction-error responses in the ACC are not indiscriminately broadcast, but are significantly excluded from a major projection target.

**Prediction-error signals in the ACC coincide with effective silencing and un-silencing epochs**

Having identified prediction-error neurons in the ACC, we next asked whether the duration of the neural prediction-error signal corresponded to the duration of causal involvement of the ACC in the task. We first asked how sustained the prediction-error response was within an individual trial. We focused on all ACC neurons with a positive prediction-error response and found that their response peaked soon after the time of expected odour onset, but remained significantly higher than both its baseline and the actual odour response until the beginning of the next trial (Fig. 5A, Wilcoxon signed-rank test prediction-error response vs baseline aligned to prediction error and next stimulus $P = 2.98 \times 10^{-15}$, $P = 3.34 \times 10^{-8}$ respectively, prediction-error response vs odour response, $P = 1.69 \times 10^{-28}$, $P = 5.45 \times 10^{-15}$ respectively). Indeed, this corresponded to the duration of the requirement of the ACC in the task, since silencing the ACC both during the ITI and peri-stimulus period on each trial (Fig. 5B) caused significant deficits in the switching behaviour, although each to a smaller degree than continuous silencing (Fig. 3C). Thus, ACC activity is required not only around the moment of expectation violation, when the largest prediction-error responses are present, but remains important until the start of the next trial as the animal updates its belief about the current rule.

Is the ACC required for multiple trials as a block transition occurs? To address this question, we first asked how sustained the prediction-error response in the ACC was across consecutive trials. We found that the prediction-error response was present only on the first prediction-error trial, and rapidly decayed to non-significant amplitudes on subsequent trials (Fig. 5D). If the ACC indeed enables block transitions through prediction-error signalling, the rapid decay of the prediction-error signal over trials would predict that the ACC is only required during the prediction-error trial, and not earlier or later. To directly test this prediction, we continuously silenced the ACC for the entire session and unsilenced it only on the first trial of a visual block when the prediction error occurs (Fig. 5E). The resulting behaviour showed no deficits in task switching, and mice rapidly switched between blocks (Fig. 5F, bottom, Supplementary Fig. 2E). Interestingly, the speed of behavioural switching in the un-silencing condition was significantly faster than controls, possibly due to enhanced ACC activity from the removal of inhibition (see also Fig. 6). Critically, after switching to the new block, the mice performed highly accurate discrimination for the rest of the block despite the ACC being continuously silenced (Supplementary Fig. 2F). The rapid task switching could not be accounted for by a startle response to the light offset, since the mice did not lick in response to the light offset, or the visual stimulus immediately

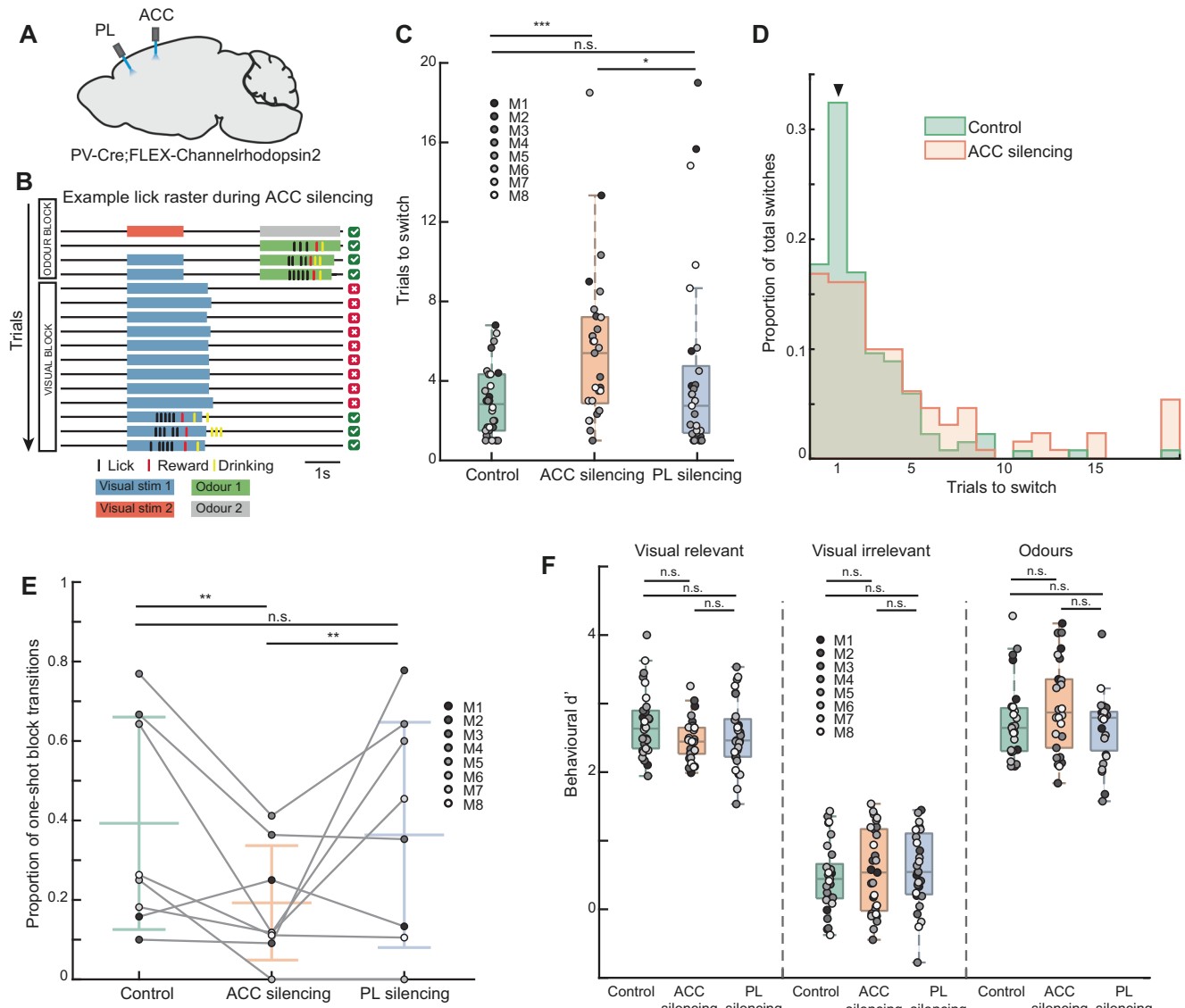

**Fig. 3 | Optogenetic inhibition of ACC impairs task-switching. A** Schematic of bilateral optic fibre implantation targeting both ACC and PL in each mouse. **B** Example lick raster showing stimuli, lick and reward times during a transition from an odour to visual block during ACC silencing leading to impaired switching behaviour. **C** Average number of trials required in a session to switch from an odour to visual block, *N* = 30 sessions, 8 mice, 3–4 sessions per silencing condition per mouse, shades of grey indicate individual mice here and below. Median ± interquartile range (IQR) for control 2.83 ± 2.83 trials, continuous ACC silencing 5.4 ± 4.34 trials, continuous PL silencing 2.75 ± 3.37 trials. Two-sided Wilcoxon signed-rank test comparing control and continuous ACC silencing *P* = 0.0006, control and continuous PRL silencing *P* = 0.78, continuous ACC silencing and continuous PRL silencing *P* = 0.02. **D** Histogram of number of trials taken to switch from odour to visual blocks during control and continuous ACC silencing sessions. *N* = 136 and 131 control and ACC silencing transitions respectively. One-shot transitions indicated by arrowhead. **E** Proportion of transitions from odour to visual blocks that were one-shot. Grey circles represent means for individual mice (*N* = 8),

lines represent means ± STD across mice. Mean ± STD for control 0.39 ± 0.27, continuous ACC silencing 0.19 ± 0.14, continuous PL silencing 0.36 ± 0.28. Two-sided Chi-squared test of proportions comparing control and continuous ACC silencing *P* = 0.001, control and continuous PL silencing *P* = 0.65, continuous ACC silencing and continuous PL silencing *P* = 0.005. **F** Steady-state stimulus discrimination performance after each successful block transition. Behavioural d-primes to visual stimuli in a visual block: control, continuous ACC silencing and continuous PL silencing sessions, 2.64 ± 0.55, 2.44 ± 0.38 and 2.46 ± 0.55, respectively, visual stimuli in an odour block 0.44 ± 0.5, 0.54 ± 1.19, and 0.54 ± 0.89, respectively, and odour stimuli 2.64 ± 0.63, 2.87 ± 1.0, and 2.79 ± 0.57, respectively. Two-sided Wilcoxon signed-rank tests as indicated, all Ps > 0.05. Each datapoint represents averages for a single session, recorded from 8 mice performing 3-4 sessions each. For all boxplots the centre mark indicates median, with the upper and lower bounds indicating 75th and 25th percentile respectively, and the whiskers indicating the most extreme datapoints not considered outliers.

following the light offset (middle row in Fig. 5E), but instead licked in response to the subsequent visual stimulus, after the odour prediction error. Thus, the ACC is specifically required only for processing the prediction error and is thereafter no longer required for accurate task performance. These results provide direct evidence for prediction-error signals in the ACC driving task-switching[2,4], and are consistent with a prominent theory suggesting that the PFC largely signals the non-occurrence of expected outcomes[52].

## Larger prediction-error signals in ACC precede successful one-shot block transitions

Finally, to determine whether prediction-error neurons in the ACC actively contribute to the behavioural transitions across blocks, we asked if the amplitude of prediction-error responses at block transitions was related to the subsequent success in behavioural switching. We divided the odour to visual block transitions into one-shot and slower than one-shot block transitions (Fig. 6A, *N* = 51% one-shot and

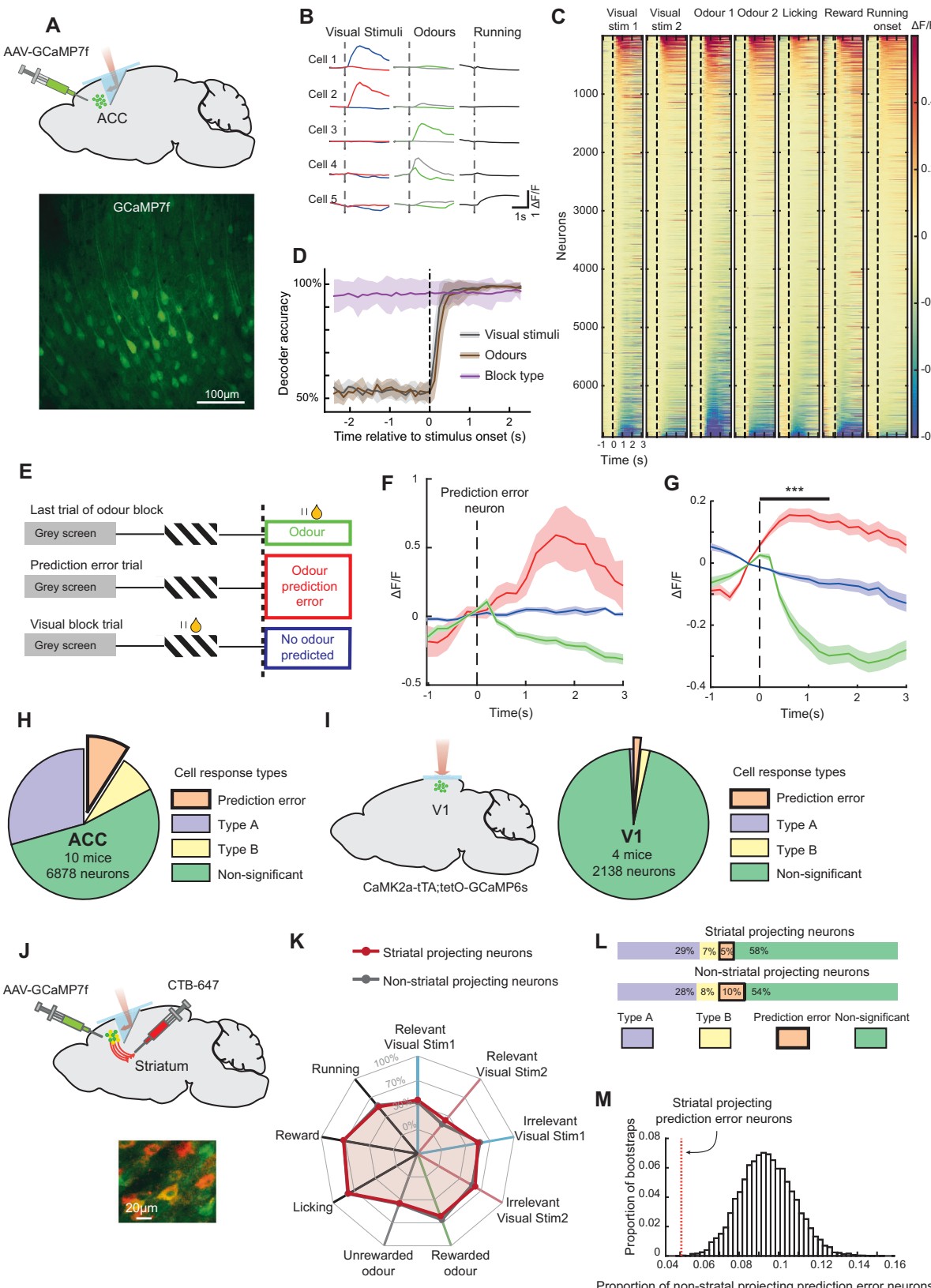

49% slower transitions). Prediction-error neurons with a positive response were more active preceding one-shot block transitions, compared to slower transitions (Fig. 6B) when aligned to the prediction-error event (Wilcoxon signed-rank test, $P = 0.0003$) and also when aligned to the next stimulus onset ($P = 0.019$, $N = 146$ neurons). This result was also true when combining all prediction-error

neurons ($P = 8.57 \times 10^{-9}$, $N = 616$ neurons). This result could not be accounted for by differences in running speed across the two conditions (Supplementary Fig. 7A). The pupil diameter also did not show significant differences between the one-shot and slower block transitions, and largely reflected running speed changes (Supplementary Fig. 7B).

**Fig. 4 | ACC neurons encode prediction-error signals. A** Top, schematic depicting two-photon imaging of neurons expressing GCaMP7f in the ACC through a microprism implant. Bottom, example ACC imaging site. **B** Responses from 5 example neurons to task stimuli and running onsets. Shading here and below indicates SEM. **C** Mean responses of all ACC neurons ($N = 6878$ neurons) aligned to task stimuli and behaviour. Activity was aligned −1 s to 3 s around task stimulus or behaviour onset and mean baseline (−0.5 s to 0 s) subtracted. Each condition is sorted by the averaged activity from 0 to 1 s. **D** Time course of decoding accuracy of a binary classifier using neuronal activity of simultaneously imaged neurons from the ACC, mean of 9 sessions from 8 mice. Grey, decoding the two visual stimuli in the visual block, brown, decoding the two odour stimuli in the odour block, and purple, decoding block-type from activity aligned to the non-rewarded visual stimulus onset in the two blocks. **E** Schematic of behaviour during a one-shot transition from an odour to visual block, indicating the three trial types used to identify prediction-error neurons. **F** Example prediction-error neuron with a significantly larger response to the odour prediction-error condition (red), compared to the actual delivery of the odour (green) or trials where odours were neither predicted nor delivered (blue). Data are presented as mean responses +/− SEM. **G** Average response of all prediction-error neurons with a positive response to the odour prediction-error condition, in the three conditions as described in **F** ($N = 168$ neurons, Two-sided Wilcoxon signed-rank test between the prediction error and odour

conditions averaged 0 to 1.5 s, ***$P = 1.69 \times 10^{-28}$). Shading indicates SEM. **H** Proportions of neurons with significantly different activity (averaged 0 to 1.5 s) between the three trial types described in **E**. Type A and Type B were significantly different only when comparing odour prediction error to odour ($N = 2007$ neurons, 29%) or to no odour predicted conditions ($N = 567$ neurons, 8%) respectively. Prediction-error neurons were defined as neurons significantly different in *both* comparisons ($N = 616$ neurons, 9%, total 6878 neurons, 10 mice). **I** Left, schematic of two-photon calcium imaging from primary visual cortex (V1). Right, same comparisons as in **H** with neurons recorded from V1 ($N = 18$ Type A, 1%, 41 Type B, 2%, and 35 prediction-error neurons, 1.6%, total 2138 neurons, 4 mice) **J** Top, schematic of retrograde labelling and imaging strategy. Bottom, example image of retrogradely labelled striatal-projecting (CTB-Alexa647 labelled) and non-striatal projecting neurons in the ACC. **K** Percentages of recorded neurons significantly responsive to each of 9 task events. **L** Proportion of prediction-error neurons from striatal-projecting and non-striatal projecting neurons in ACC. Striatal projecting: $N = 124$ Type A (29%), 30 Type B (7%), 21 prediction-error neurons (5%), total 421 neurons. Non-striatal projecting: $N = 1384$ Type A (28%), 392 Type B (8%), 484 prediction-error neurons (10%), total 4888 neurons, 8 mice). **M** Bootstrapped distribution of proportion of prediction-error neurons in non-striatal projecting neurons. Data for proportion of striatal projecting prediction-error neurons indicated as vertical line. Data is outside the 99% confidence intervals.

Since our belief-state RL model of the behaviour included an explicit context-prediction-error signal, we asked if this signal also predicted one-shot block transitions across contexts. The model revealed a similar pattern, where the amplitude of the noisy prediction-error signal was predictive of future one-shot transitions (Fig. 6C, Wilcoxon rank sum test, $P < 10^{-8}$, $N = 70$ transitions). These results support the claim that prediction-error neurons in the ACC play a key role in driving rapid task-switching behaviour, particularly when they produce larger prediction-error signals.

## VIP interneurons contribute to prediction-error computation in the ACC

Local inhibitory circuits are crucial in shaping cortical activity[53], and are believed to be necessary for computing prediction errors[2]. These circuits need to compare predictions and observations, with a mismatch between the two leading to prediction-error signals. While a diversity of inhibitory circuits may compute prediction-errors[36], VIP interneuron disinhibition[37–39] is hypothesized to be key in this process[36] (Fig. 7A). Indeed, VIP driven disinhibition interacts with a perceptual prediction error during visuo-motor coupling mismatch in V1[54]. We asked whether VIP interneurons played a role in producing cognitive prediction-error responses in the ACC in our task. We employed an all-optical approach, where we photoactivated or photoinhibited VIP interneurons while simultaneously measuring the activity of VIP and non-VIP neurons in the same brain region with in-vivo two-photon calcium imaging (Fig. 7A, B). Importantly, the VIP perturbations were in one hemisphere only, and did not lead to any changes in the task-switching behaviour itself (Wilcoxon signed-rank test comparing behavioural d′ for the relevant visual stimuli, irrelevant visual stimuli, and odours between control and ITI VIP activation or inhibition sessions, all Ps > 0.05, similar Wilcoxon signed-rank test comparing the number of trials to switch between blocks all Ps > 0.05).

On photoactivating VIP cells in the ACC in passive mice with increasing light powers (Fig. 7C bottom, inset), we observed monotonically increasing population activity in non-VIP cells (Fig. 7C). Since non-VIP cells in cortex are predominantly excitatory pyramidal neurons, this experiment demonstrated the effective disinhibition induced by our all-optical approach. We next photoactivated VIP cells in the ACC as mice performed the task-switching behaviour. VIP cells were photoactivated on each trial only during the inter-trial-interval (ITI) period, which encompassed the prediction-error event at block transitions (Fig. 7D). We compared the proportion of prediction-error neurons in the VIP photoactivation sessions to control sessions and found that VIP activation during the ITI period strongly reduced the

percentage of prediction-error neurons from 10% to 2% (Fig. 7E, Chi-squared test of proportion $P < 0.0001$). To determine how VIP photoactivation affected the response of ACC neurons to prediction-errors, we measured the average response amplitude to the odour prediction-error event, and odour stimulus, from positively responding prediction-error neurons (Fig. 7F). We found that VIP photoactivation led to a significant reduction only in the response to the odour prediction-error, demonstrating that VIP photoactivation limits the prediction-error response amplitude (Fig. 7F bottom). Critically, we confirmed that our VIP activation did not lead to widespread deficits in ACC responses, since our perturbation did not affect representations of stimuli and block rules. A binary classifier as used in Fig. 4D was able to accurately decode the identity of visual and odour stimuli after stimulus onsets and the block type before and after stimulus onsets, both when VIP cells were photoactivated during the visual stimulus onsets (Fig. 7G) and during the ITI period (data not shown). Thus, inducing disinhibition through VIP interneuron activation specifically and heavily disrupted prediction error signalling in the ACC.

Finally, to prove their necessity in generating prediction error responses in the ACC, we optogenetically silenced VIP interneurons during the task. VIP inhibition during the ITI period strongly reduced the percentage of prediction-error neurons from 5% to 0.6% (Fig. 7H, Chi-squared test of proportion $P < 0.0001$). Again, a binary classifier was able to accurately decode visual and odour stimuli and block type, both when VIP cells were photoinhibited during the visual stimulus onsets (Fig. 7I) and during the ITI period (data not shown). Crucially, light-only control mice which did not express any opsin showed no change in the proportion of prediction-error neurons detected on presentation of the same optogenetic light (Figs. 7J, 3.6% to 4.3%, Chi-squared test of proportion $P = 0.46$). Together, these results demonstrate that bidirectional perturbation of VIP interneurons strongly and specifically disrupts prediction-error computations in the ACC. Future work may test the role of other cell classes such as PV interneurons, and the sources of prediction and observation signals in this computation (Fig. 7A).

## Discussion

In this study, we demonstrate that mice can perform rapid and repeated block transitions in a cross-modal task switching paradigm. A theoretical RL model suggested that this behaviour was driven by a cognitive prediction-error signal, and in agreement with this prediction, we identified a neural signal which represented a cognitive prediction error in the ACC. Silencing and un-silencing the ACC in precise

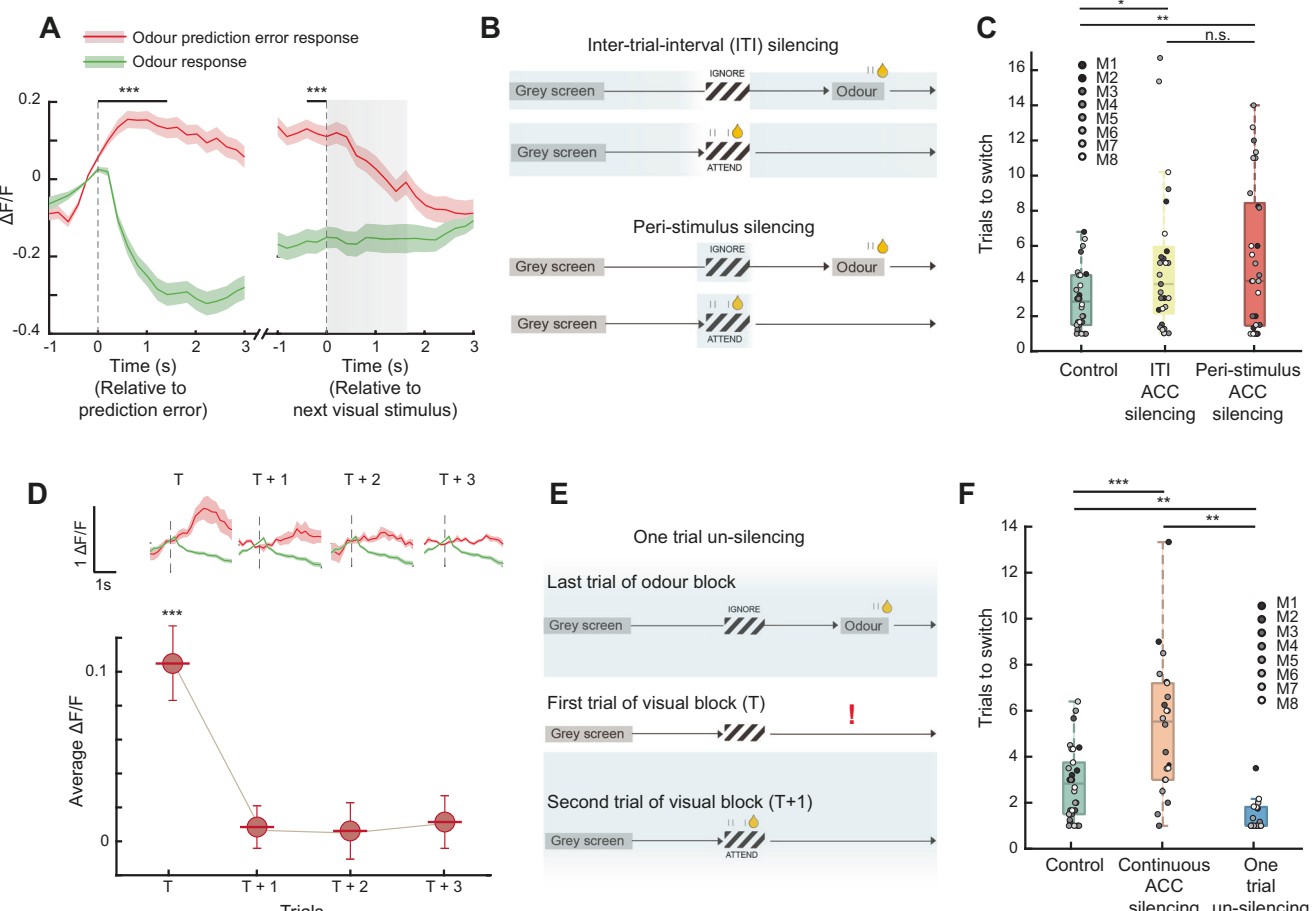

**Fig. 5 | ACC prediction-error signal duration corresponds to requirement of the ACC in task switching. A** Mean response of all positively responding prediction-error neurons ($N = 168$ neurons) aligned to the odour prediction-error event, and to the next visual stimulus onset (grey shading). Response shown to the odour prediction-error event (red) and actual odour delivery (green). Two-sided Wilcoxon signed-rank test between the two conditions averaged 0 to 1.5 s from prediction-error event and −0.5 to 0 s from the next stimulus onset, ***$P = 1.69 \times 10^{-28}$ and $P = 5.45 \times 10^{-15}$ respectively. **B** Schematics depicting inter-trial-interval (ITI) and peri-stimulus silencing epochs. **C** Number of trials required in a session to switch from an odour to visual block, median ± IQR here and below, control 2.83 ± 2.83 trials, ITI ACC silencing 3.8 ± 3.79 trials, and peri-stimulus ACC silencing 4.0 ± 6.98 trials. Two-sided Wilcoxon signed-rank tests comparing control and ITI sessions $P = 0.02$, control and peri-stimulus sessions $P = 0.009$, ITI and peri-stimulus sessions $P = 0.39$. Each datapoint represents averages from a single session, recorded from 8 mice performing 3-4 sessions each. **D** Mean odour prediction-error response relative to pre-event baseline aligned to the same temporal epoch over consecutive trials (T is first prediction-error event in an odour to visual block transition), shown

up to 3 trials following first prediction-error event (two-sided Wilcoxon signed-rank test, average −0.5 to 0 s compared to 0 to 1.5 s, ***$P = 2.98 \times 10^{-15}$). Each datapoint represents the mean response across all positively responding prediction-error neurons, with error-bars indicating SEM. Top, example neuron responses showing the rapid decay of prediction-error response over trials, mean responses +/− SEM. **E** Schematic depicting silencing during the entire session except for one trial at each odour to visual block transition. **F** Number of trials required in a session to switch from an odour to visual block, control 2.83 ± 2.25 trials, continuous ACC silencing 5.4 ± 4.34 trials, one trial un-silencing 1.0 ± 0.82 trials, two-sided Wilcoxon signed rank test comparing control and continuous ACC silencing sessions $P = 0.0006$, control and one trial un-silencing $P = 0.0084$, continuous ACC silencing and one trial un-silencing $P = 0.0016$. Each datapoint represents averages from a single session, recorded from 8 mice performing 2–3 sessions each. For all boxplots the centre mark indicates median, with the upper and lower bounds indicating 75th and 25th percentile respectively, and the whiskers indicating the most extreme datapoints not considered outliers.

time windows demonstrated that this neural prediction-error signal was causally required for the mouse to update its behaviour between blocks. The amplitude of the prediction-error signal preceding a trial was related to whether or not a mouse would correctly switch its behaviour in the subsequent trial. Finally, VIP interneurons were identified as a key inhibitory cell class in the computation of cognitive prediction errors. Taken together, our results provide causal and mechanistic evidence for a longstanding idea, that the ACC computes cognitive prediction errors to guide flexible behaviour.

By training mice to perform repeated one-shot block transitions, we encapsulated a complete cognitive switch in a well-defined time window of a few seconds. Such rapid and complete cognitive transitions are akin to 'Aha! moments', where some information or insight allows an abrupt updating of one's internal model of the world[55]. These

rapid cognitive transitions are difficult to study experimentally – one-shot transitions have primarily been observed in primates[34,35], and previous studies in rodents required tens to hundreds of trials to complete a behavioural transition[12,26–32,42]. Periods of intermediate accuracy which accompany these slower transitions prevent unambiguously relating a neural signal to its behavioural consequence. In our task, a well-defined prediction error leads to a demonstrable update in the animal's cognitive model in the next few seconds. Moreover, when studying the moment of cognitive transition, multiple block transitions are required in a single session for sufficient statistical power. In addition, to be certain of an animal's internal model of the world requires that the animal performs at high accuracy in the given task before the block transition, typically resulting in longer and fewer blocks. Our task provided an unprecedented number of

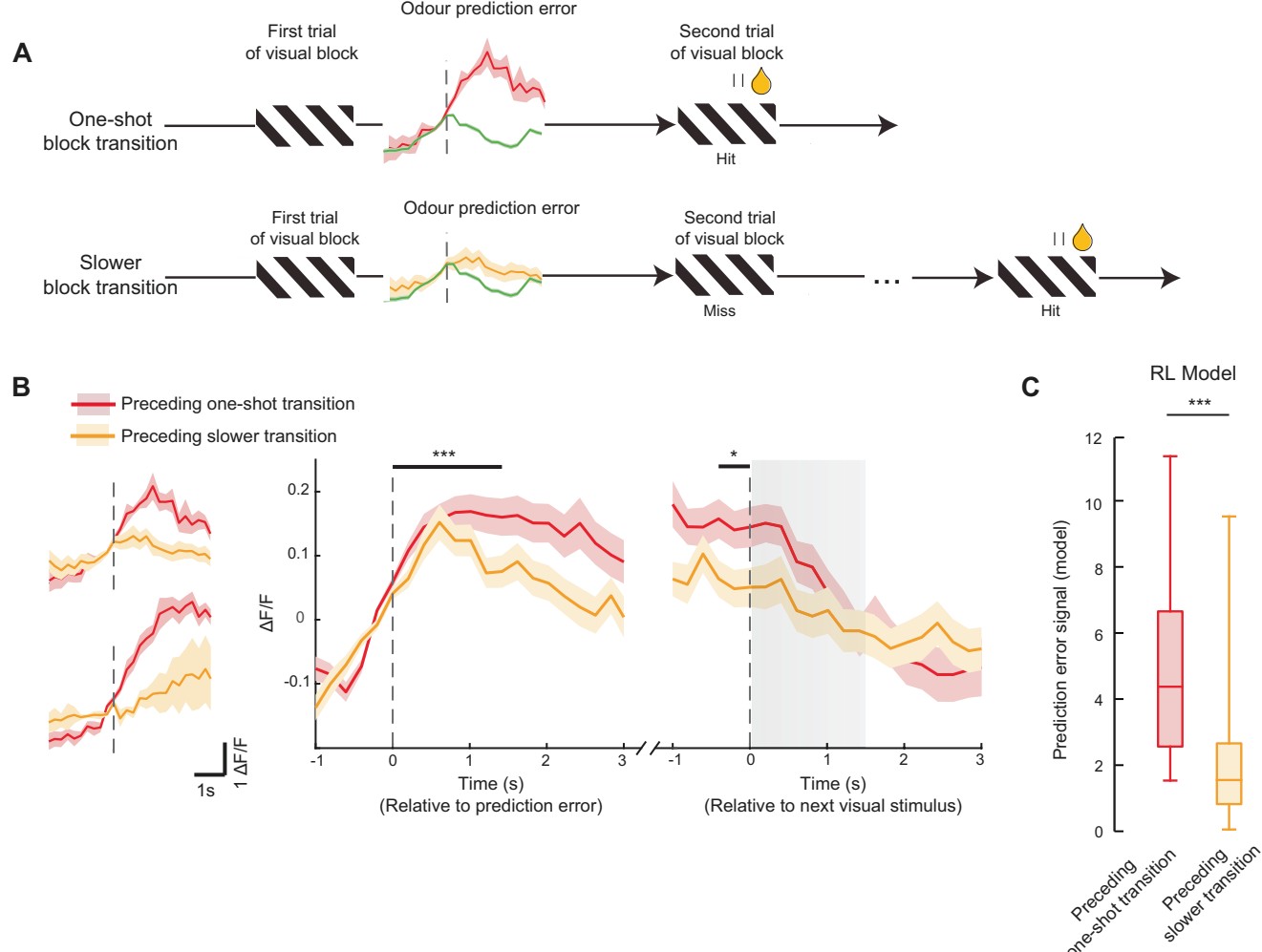

**Fig. 6 | Larger prediction-error signals in ACC precede one-shot block transition behaviour. A** Schematic showing odour to visual block transitions and the prediction-error response preceding two scenarios, top, where the mouse executes a one-shot block transition, or switch, and bottom, where the block transition is slower than one-shot. Red and yellow responses are prediction-error responses from an example neuron, green shows the response to delivery of odour, shading indicates SEM here and below. **B** Left, odour prediction-error response of two example neurons preceding a one-shot (red) or slower transition (yellow). Right, mean responses across all positively-responding prediction-error neurons ($N = 146$

neurons) preceding one-shot and slower transitions (two-sided Wilcoxon signed rank test between the two conditions averaged 0 to 1.5 s from prediction-error event and −0.5 to 0 s from the next stimulus onset, ***$P = 0.0003$, *$P = 0.019$). Shading indicates SEM. **C** Prediction-error signal amplitude from the Belief-State RL model, preceding one-shot transitions and slower transitions, ***$P < 10^{-8}$, $N = 70$ transitions. For the boxplots the centre mark indicates median, with the upper and lower bounds indicating 75th and 25th percentile respectively, and the whiskers indicating the most extreme datapoints not considered outliers.

repeated, highly rapid block transitions with mice switching between high accuracy behaviours, which was crucial for conclusively assigning a behaviourally relevant role to the neural prediction-error signal.

Cognitive prediction errors are distinct from prediction-error signals in other domains, such as reward prediction errors in dopaminergic neurons, sensory prediction errors in sensory cortex, or sensorimotor prediction errors in the cerebellum[2,56–59]. Cognitive prediction errors enable flexible cognition, where one needs to maintain an abstract cognitive context or model of the world in mind, which is updated when a prediction from the cognitive context or model is violated. This is a distinct problem to solve compared to other prediction-error computations and involves different brain regions[9]. Critically, the prediction-error responses we found in this study did not resemble classical negative reward signals such as those found in dopaminergic neurons. Instead, this signal resembles an outcome prediction error[52] which relates to errors in predicting outcomes based on the currently believed rules of the task.

The computation of a prediction-error signal in our task requires comparing an internally generated prediction signal with an external

odour signal. The origin of the odour prediction signal is currently unclear, and may reach the ACC from other prefrontal areas[11,60–62]. However, the ACC itself robustly represents the ongoing task rules as well as external stimuli, and thus may compute the prediction error autonomously.

An important goal in understanding the neural circuit basis of cognition is to identify the circuit which compares predictions with observations. While this circuit has been studied in other contexts, such as for the dopaminergic reward-prediction system[57] or the visuomotor mismatch system[56], it is poorly understood for cognitive rule-updating. In this study, we took advantage of a temporally well-defined cognitive prediction-error signal to take the first steps in uncovering the circuit involved. We found VIP-driven disinhibition to be key, and expect future studies to reveal further details about the role of other inhibitory cell classes in this circuit.

The recurrently connected nature of cortical circuits may suggest that perturbation of any cell class will invariably lead to a disruption of processing in that region. However, VIP modulation can be orthogonal to cognitive modulations in other cortical regions[63], and importantly,

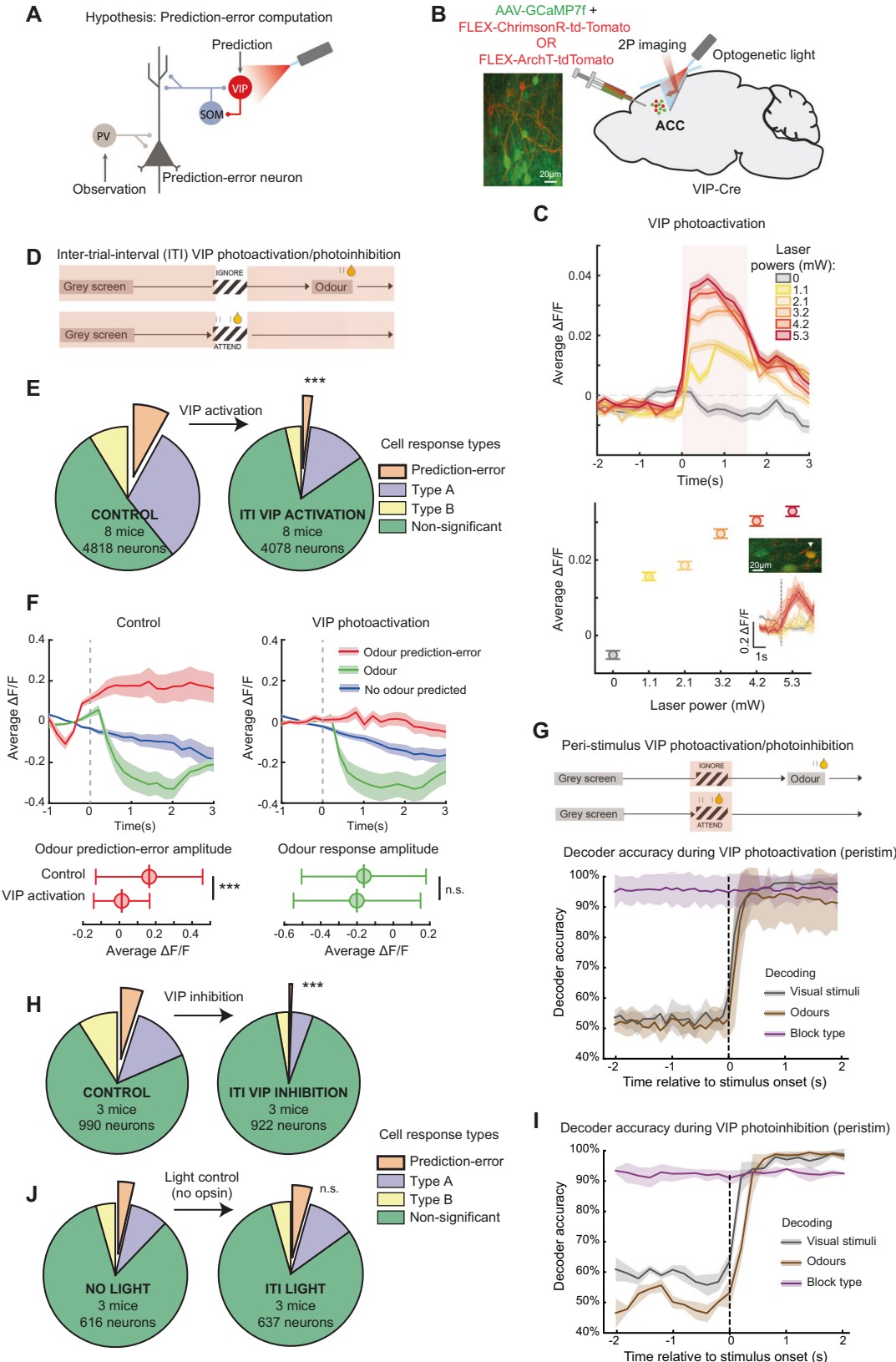

here we show that VIP perturbations in the ACC did not affect stimulus and context representations, while disrupting prediction-error signals.

Other cortical association areas may play a role in task switching, such as orbitofrontal cortex[12,27] and posterior parietal cortex[64], as well as subcortical areas including higher-order thalamic nuclei[65]. Future

work understanding the role of these and other regions, would need to establish both the nature of the prediction-error signal and its requirement in behaviour. Although we have ruled out a role for PL in our task, earlier studies have shown that PL circuits are involved in flexible behaviour[45,66]. However, these studies investigated slower

**Fig. 7 | All-optical VIP interneuron bidirectional perturbations in ACC disrupt prediction-error signalling. A** Hypothesis from theoretical work for a cortical circuit to compute prediction errors. The prediction-error neuron will respond (positive or negative responses) when predictions and observations do not cancel each other out. Part of this hypothesis is tested by activating and inhibiting VIP interneurons during task-switching. **B** Left, example site with neurons expressing GCaMP7f in green and VIP interneurons expressing the excitatory opsin ChrimsonR in red. Right, schematic of our all-optical approach: simultaneous 2-photon imaging and optogenetic activation or inhibition of VIP interneurons in the ACC. **C** Top, peri-stimulus time histograms of mean activity of all non-VIP neurons ($N = 2467$ neurons, 8 mice) aligned to light onset, at 5 different light powers, for 1.5 s. Shading and error bars indicate mean ± SEM across all neurons from all 8 mice here and below. Bottom, average activity of all neurons across light presentation window (0–1.5 s). Inset: Image of an example VIP interneuron (arrowhead) and its response to increasing light powers, demonstrating photoactivation. **D** Schematic of inter-trial-interval (ITI) activation epoch. **E** Proportions of cells showing prediction-error responses and other cell classes as in Fig. 4H. Left, control and right, ITI VIP activation sessions from

the same sites, performed on subsequent days. **F** Top, average activity in each trial type of positively responding prediction-error cells identified in all three sessions ($N = 38$). Shading indicates SEM. Bottom, mean response (0 to 1.5 s) of these cells to odour prediction error (left, red) and odour (right, green). Two-sided Wilcoxon rank-sum tests comparing control vs ITI sessions, prediction-error responses $P = 1.1 \times 10^{-4}$, odour responses $P = 0.93$, with whiskers indicating SEM. **G** Top, schematic of peri-stimulus activation/inhibition epoch. Bottom, time course of decoding accuracy of a binary classifier using neuronal activity during peri-stimulus VIP activation sessions ($N = 4603$ neurons, 8 mice), decoding task stimuli as in Fig. 4D. Shading indicates SD across mice. **H** Proportions of cells showing prediction-error responses and other cell classes as in Fig. 4H. Left, control and right, ITI VIP inhibition sessions from the same sites, performed on subsequent days. **I** Time course of decoding accuracy with a binary classifier using activity during peri-stimulus VIP inhibition sessions ($N = 888$ neurons, 3 mice). Shading indicates SD across mice. **J** Proportions of cells showing prediction-error responses and other cell classes as in Fig. 4H. Left, no light and right, ITI light only control sessions from the same sites, performed on subsequent days.

---

forms of behavioural transitions, compared to our largely one-shot transitions, possibly accounting for the difference with our results. Indeed, there are multiple types of surprise signals which may be processed differently by different brain regions[9,67].

Where is the prediction-error signal sent to in order to drive the behavioural changes? We have found that the striatum is an unlikely candidate, despite being a major projection target of the ACC, since the prediction-error signal is under-represented in ACC neurons projecting to the striatum. Other likely brain regions may include the locus coeruleus, which has been implicated in updating current strategies through norepinephrine signalling[17,18,68,69], or the dorsal raphe nucleus, which promotes behavioural flexibility through serotonin and glutamate signalling[70,71].

Disruption of cognitive flexibility in humans may lead to excessive transitioning between attentional states in ADHD, or excessive persistence in one or a few repetitive behaviours in ASD[72]. However, the neural circuit basis of these conditions is poorly understood. Our results provide a crucial insight regarding the role of the ACC in transitioning between, rather than maintaining, cognitive states. Thus, atypical ACC activity patterns may contribute to excessive or insufficient cognitive transitions in humans, and a more detailed understanding of how ACC circuits produce and transmit prediction errors may provide insights to better understand these conditions.

## Methods

All experimental procedures were carried out in accordance with the institutional animal welfare guidelines and licensed by the UK Home Office.

### Animals and surgical procedures

For all surgeries, mice were anaesthetised using isoflurane, at 4% concentration for induction and at 0.5–1% for maintenance. Additional drugs were used to provide analgesia (Metacam 5 mg/kg), anti-inflammatory effects (dexamethasone 3.8 mg/kg), and to reduce mucus secretions (Atropine 0.08 mg/kg). Eye-cream (Maxitrol) was applied to the eyes to prevent drying and body temperature was maintained at 37 °C using a heating mat and rectal temperature probe (Harvard Apparatus). Injections of antibiotic (Betamox 120 mg/kg) and analgesia (methadone hydrochloride 10 mg/kg) were given before the withdrawal of anaesthesia, and further analgesia was given daily for 1–2 days during recovery of the animal.

For the two-photon imaging experiments, 10 VIP-Cre mice (C57BL/6-VIP$^{tm1(cre)zjh}$, The Jackson Laboratory, strain # 010908, P42-49, 6 males and 4 females) were used. An additional 4 mice of the same genotype (1 male, 3 females) were used together with the imaging mice for behavioural analysis. A circular piece of scalp was removed and the underlying skull was cleaned. Small holes were drilled in the skull

above injection sites, located using stereotaxic co-ordinates. Injections of a mixture of viruses expressing GCaMP7f (pGP-AAV9-syn-jGCaMP7f-WPRE, Addgene) and Cre-dependent ChrimsonR (pAAV5-syn-FLEX-rc[ChrimsonR-tdTomato], Addgene) were made in the anterior cingulate cortex (ACC, +0.9 mm AP, −1.3 rising to −0.8 mm DV) of either the left or right hemisphere (+/− 0.55 mm ML), using glass pipettes and a pressure micro-injection system (Picospritzer III, Parker). An injection of Cholera Toxin Subunit B (recombinant) conjugated to Alexa Fluor 647 (ThermoFisher) was made in the striatum of the same hemisphere ( +1.2 mm AP, +/− 1.5 mm ML, −2.5 rising to −2.0 mm DV) to retrogradely label cells projecting to the striatum. 8 out of the 10 mice had good enough quality z-stacks in the ACC to visualise CTB-Alexa647 and were used to identify striatal projecting ACC neurons. 8 out of the 10 mice exhibited sufficiently consistent behaviour to study the effects of VIP photoactivation. For the VIP silencing experiments (Fig. 7H) 3 VIP-Cre mice were injected with a mixture of viruses expressing GcaMP7f (pGP-AAV9-syn-jGCaMP7f-WPRE, Addgene) and Cre-dependent ArchT (pAAV-FLEX-ArchT-tdTomato (AAV5), Addgene). For the light-only control experiments (Fig. 7J), an additional 3 VIP-Cre mice were injected with a mixture of viruses expressing GCaMP7f and Cre-dependent tdTomato (pAAV-FLEX-tdTomato (AAV1), Addgene).

A circular craniotomy (diameter = 3 mm) was made above the ACC imaging site, with a centre 300 μm posterior to ACC injections. A $1.5 \times 1.5 \times 1.5$ mm right-angled microprism with a reflective hypotenuse (Tower Optical) fixed to a glass coverslip using ultraviolet light-cured glue (Thorlabs) was slowly lowered into the craniotomy, with the vertical face closest to the injection site. The glass coverslip was fixed in place using cyano-acrylic glue (Loctite) and a custom machined aluminium head-plate was cemented onto the skull using dental cement (C&B Superbond). Imaging and behavioural training started approximately three weeks after surgery.

For the widefield imaging experiments, four male wildtype mice (C57BL/6;129-Nrxn1$^{tm1Sud}$) from the ages of 21 to 27 weeks were used (all mice were wildtype C57BL/6). At least four weeks before surgery mice were given an intravenous injection of AAV PHP.eB GCaMP7f (Zurich vector core). For the surgery, the skin overlying the skull was removed and the edges of the skin were secured with tissue adhesive (Vetbond, 3 M). The overlying connective tissue on the skull was removed and a layer of transparent dental cement (C&B Superbond) was applied to cover all exposed skull and to secure a custom aluminium headplate. Following this 5 layers of 2.5ul of cyanoacrylate glue (Zap-A-Gap CA + , Pacer Technology) were thinly applied onto the cement to increase the transparency. After recovery from surgery for at least 5 days mice began habituation and behavioural training.

For the optogenetic silencing experiments, 8 transgenic mice (P42-49, 4 males and 4 females) expressing Channelrhodopsin-2 in

parvalbumin-expressing interneurons were used, obtained by crossing FLEX ChR2 mice (Gt(ROSA)26Sor[tm32(CAG-COP4*H134R/EYFP)Hze]) and PV-Cre mice (Pvalb[tm1(cre)Arbr], The Jackson Laboratory, strain # 204109 and # 017320 respectively). Two small holes were drilled above the ACC and prelimbic cortex (PL) of each hemisphere. Dual-core cannulae with bilateral optical fibres (Thorlabs), each with a diameter of 200 μm and 0.39NA, cut to a length of <3 mm, were implanted in the ACC (+ 0.9 mm AP, −1.2 mm DV, +/−0.35 mm ML) and PL (+ 2.6 mm AP,−1.25 mm DV, +/− 0.35 mm ML), and the stainless steel ferrules were bonded to the skull using dental cement (C&B Superbond), along with a custom machined head-plate. PL implants were inserted at a 25° angle (relative to vertical) through holes drilled 0.8 mm anterior to PL. We performed light-only control experiments by comparing 3 PV-Cre mice (1 male and 2 females) with 3 PV-Cre crossed with FLEX-ChR2 mice (1 male and 2 females), which were implanted with optical fibre cannulae above ACC only. After recovery from surgery for at least 5 days mice began habituation and behavioural training. Mice were housed in a reversed-light-cycle cabinet illuminated between 7 pm and 7am, maintained at a temperature of 22 °C and 56% humidity.

## Imaging

Two-photon imaging was performed using a custom-built resonant scanning two-photon microscope (Cosys) and a Chameleon Vision S laser (Coherent) at 930 nm using a 16X, .8NA objective (Nikon). Images were acquired using a 12 KHz resonant scanner (Cambridge Technology) and an FPGA module (PXIe-7965R FlexRIO, National Instruments). Two-photon calcium imaging of GCaMP7f-labelled neurons in the ACC was performed across 40 training sessions and 48 full task-switching sessions in these 10 mice. The microprism depth, injection coordinates and cell morphology indicated that the imaging sites were largely located in layer 5. Multi-plane imaging was performed using a piezo-electric objective scanner (Physik Instrumente). Depending on the depth of GCaMP7f expression, each imaging volume consisted of either 6 or 8 imaging planes, 40 μm apart, giving an effective imaging rate of 6.4 or 4.8 Hz per volume respectively.

Mice were trained first in the visual discrimination task, then had at least 3 training sessions in the visual-odour block switching task. Once mice had learned the switching task, at least 3 recordings of the mice performing the task were taken per mouse. Before each recording session the same imaging site was found by matching anatomical landmarks.

After all in-vivo imaging data had been collected, a final high-quality image stack was acquired under anaesthesia. Subcutaneous injections of ketamine (100 mg/kg) and xylazine (16 mg/kg) were used to induce anaesthesia, with further injections of ketamine to maintain anaesthesia if necessary. Eye-cream (Maxitrol) was used to prevent drying, and body temperature was maintained using a heating pad.

Widefield imaging was performed on a custom built inverted tandem lens macroscope (Cosys), with two photographic lenses (AF-S NIKKOR 85 mm f/1.8 G lens and AF NIKKOR 50 mm f/1.4D Lens). The brain was illuminated with interleaved collimated blue (470 nm, Thorlabs M470L4) and violet light (405 nm, Thorlabs M405L4) at an irradiance of -0.03 mW/mm². Images were recorded with a CMOS camera (Point Grey Research Grasshopper3) at frame rate of 54 Hz. LEDs and camera frame acquisition were triggered using a digital microprocessor (Teensy 3.2).

Widefield data was pre-processed using the methods described in ref. 73. The widefield video underwent motion correction and the brain images were aligned within and across mice by manual rigid alignment to a number of anatomical landmarks. The video data was compressed and denoised by performing SVD on the matrix of pixels × time and retaining the top 500 components. The ΔF/F was computed for each pixel by taking the difference between F and F0, and dividing by F0, where F0 was the mean value across the entire session. Traces were filtered with a 0.0033 Hz high-pass second order Butterworth filter,

and an additional 7 Hz lowpass filter was applied to the violet illumination trace. To correct for haemodynamic artefacts, a scaled version of the violet illumination trace was subtracted from the blue illumination trace for each pixel. This scaling factor was found by regressing the violet trace onto the blue trace. To account for overt movement-related brain activity, we fit a ridge regression model to the data, predicting brain activity from a number of movement regressors. These included a binarized lick trace, with lags up to 500 ms as well as instantaneous running speed and average face motion energy. Running speed and face motion energy were divided by twice their standard deviation to ensure all regressors had approximately the same scale and were penalised equally by ridge regularisation. Ridge penalties were selected using fivefold cross-validation from 36 values spaced logarithmically between $10^{-2}$ and $10^5$ selecting the ridge penalty which resulted in the lowest cross validated mean squared error. Penalties were selected independently for each pixel. We then subtracted this predicted activity and all subsequent analysis was performed on the model residuals.

## Behavioural training

The behaviour apparatus and training were similar to previous studies[74,75]. Mice were trained on a visual discrimination task for up to two weeks, until discrimination performance reached threshold, before training them on the switching task (see below). Mice had free access to water but were food restricted to maintain at least 85% of their free-feeding body weight (typically 85–90%, 2–3 g of standard food pellets per animal per day). A reward delivery spout was positioned near the snout of the mouse, and licks to this spout were detected using a piezo disc sensor and custom electronics. The reward was a 10% solution of soy milk powder (SMA Wysoy) delivered by opening a pinch valve (NResearch) controlled through custom electronics. The mouse's running speed on the cylinder was measured using an incremental rotary encoder (Kübler). Two luminance-corrected monitors (luminance metre LS-100, Konica Minolta) positioned at 45° angles and 25 cm distance to the mouse delivered visual stimuli.

Animals were habituated to handling and gentle restraint over two to 3 days, before they were head-fixed and trained to run on a poly-styrene cylinder (20 cm diameter) for one to four days. This period was also used to find suitable imaging sites. After the habituation phase, mice performed one behaviour session in which the movement of the gratings was linked to the mouse's movement on the wheel. Subsequently, mice were trained to self-initiate trials by sustained running on the wheel for at least 2.8 s and an added random duration drawn from an exponential distribution with mean 0.4 s (trial structure and all timings shown in Supplementary Fig. 1A). At this point one of two drifting sinusoidal visual gratings were randomly presented, drifting in the opposite direction to the direction of running, with a fixed spatial and temporal frequency of 0.1 cycles per degree and 2 Hz respectively. The rewarded and unrewarded gratings were oriented +/−20° relative to vertical, symmetrically on both screens. When the rewarded grating was displayed the mouse could trigger the delivery of a reward, a drop of soya milk, by licking the spout during the 'reward period', lasting from 1.0 s after the appearance of the grating to its disappearance, maximum 0.8 s into the reward period. This was recorded as a 'hit'. In some sessions, the duration of the reward period started at 1.5 s and lasted up to 1.53 s, no difference in behaviour was observed with this minor change in timings. The visual stimulus stayed on for an additional 0.8 s after reward onset while the mouse consumed the reward. If the mouse did not lick during this period, the trial was recorded as a 'miss', and a drop of soy milk was delivered shortly before the disappearance of the grating. When the unrewarded grating was presented, a single lick or more at any time until the stimulus disappearance was recorded as a 'false alarm', triggering a time-out period of 4 s in which the unrewarded grating remained on screen, and

any further licks restarted the time-out. During early training the probability of unrewarded trials was occasionally increased transiently up to 0.8 to discourage erroneous licking. All mice learned the visual discrimination task in 5–10 days, with post-learning defined as three consecutive days of discrimination with a behavioural d-prime score of 2 or above. Behavioural d-prime was calculated as: $bd' = \Phi^{-1}(H) - \Phi^{-1}(F)$, where $\Phi^{-1}$ is the normal inverse cumulative distribution function, H is the rate of hit trials, and F is the rate of false alarm trials.

Once mice had learned the visual discrimination task, they were trained in odour discrimination. After the same randomised period of sustained running, one of two odour stimuli were presented to the mouse via a polyethylene tubing positioned above the snout of the mouse. Odours were delivered through a custom-built flow dilution olfactometer calibrated with a mini PID (Aurora) at 10–20% saturated vapour concentration of two solutions, 10% soy milk (rewarded odour) and 10% soy milk with 0.1% p-Cymene mixture (unrewarded odour). The odour task structure was identical to the visual task.

Once animals were discriminating the odours accurately (typically after 30–40 trials), they were trained to switch between blocks of the olfactory and visual discrimination task. Mice typically learned to switch successfully in 1–3 days. In the olfactory blocks, 70% of odour stimuli were preceded with one of the two same visual gratings featured in the visual discrimination task (fixed duration of 1.8 s, with an identical onset delay distribution as in the visual block). In this instance neither grating was rewarded or punished, and mice learned to ignore these irrelevant gratings while accurately discriminating the odours, which were presented after the irrelevant visual grating (delay between visual grating offset and odour onset 1.8 s (in some two-photon imaging sessions this delay was 1.8 s plus an added random duration drawn from an exponential distribution with mean 0.2 s). In initial switching training sessions, a reward was delivered at the end of a rewarded grating in a visual block if the mouse had failed to lick, giving a clear indication that the grating was now relevant. By the end of early training, and for all data in this paper except Supplementary Fig. 2F, this feature was removed, requiring mice to switch between blocks through noticing unexpected stimuli alone. Block switches occurred automatically when a mouse had demonstrated a > 80% discrimination performance to the relevant stimuli (visual gratings in visual block, odours in odour block) over the last 30 trials of a block. Additionally in odour blocks mice were required to have successfully ignored all irrelevant visual gratings over the previous 10 trials before a block transition was triggered. Blocks typically contained 30 to 40 trials. Mice were deemed to have learned the switching task when they could complete sessions at these parameters with at least 3 repeats of each block type.

In order to compare the speed of behavioural switching between blocks, we applied a transition period immediately after a block transition where visual stimuli were selected according to the following rules. For odour to visual block transitions, in the first trials of a visual block only the rewarded grating (visual stim 1) was shown. When a mouse responded correctly by licking to these grating stimuli on three consecutive trials this transition period ended and the block continued with the normal 50% probabilities of visual grating identities. In the visual to odour block transitions, we applied two variations of the transition period. In the first variation, the first irrelevant visual stimulus was the otherwise unrewarded visual stimulus 2, ensuring that that the block transition was indicated by the unexpected appearance of an odour, rather than a reward prediction error. The subsequent irrelevant stimuli were visual stimulus 1, and the transition period ended when a mouse responded correctly by not licking to these irrelevant visual stimuli on three consecutive trials. Odour stimuli selection itself was kept random. In the second variation, even the first irrelevant visual stimulus was visual stimulus 1, and the subsequent rules were the same. We confirmed that mice switched equally fast in both variants of the task (Wilcoxon rank sum test, $P > 0.05$). These

transition periods were used in all behaviour sessions except muscimol silencing experiments (Supplementary Fig. 2C) and light-only controls (Supplementary Fig. 2B), in which case either visual stimulus was presented from the start of the block with 50% probability.

## Reinforcement learning model

We modelled the experimental protocol of stimuli, and rewards based on mouse actions, as a reinforcement learning environment. The environment was written in Python with the OpenAI Gym interface for ease of use with other agents. Code for, and experimental data used to train the models is available at https://github.com/adityagilra/BeliefStateRL. The environment had 5 states, two for visual cues, two for olfactory cues and an end of trial cue, and two possible actions that the agent could perform, lick and no-lick. Each trial comprised a number of steps.

In a visual block, each trial had 2 steps: in step 1, a needless lick (to the previous end of trial) was punished (−1 assuming an internal cost), and one of 2 visual stimuli were shown; in step 2: a lick led to reward (+1 corresponding to soy milk drop) if visual cue 1 was presented, or punishment (−1 corresponding to experimental timeout) if visual cue 2 was presented, and the end of trial was indicated. In an olfactory block, a trial had 2 time steps in 30% of the trials, corresponding to trials without irrelevant visual stimuli, and 3 time steps in 70% of the trials, corresponding to trials with irrelevant visual stimuli. In the 2 time steps case, in step 1, a needless lick was punished (−1), and one of the 2 odour stimuli were given. In step 2, a lick led to reward (+1) if odour 1 was delivered, or punishment (−1) if odour 2 was delivered in step 1, and end of trial was indicated. In the 3 time step case, in step 1, a needless lick was punished (−1), and one of the 2 visual stimuli were shown. In step 2, a needless lick was punished (−1), and one of the 2 odour stimuli were delivered. In step 3, a lick led to reward (+1) if odour 1 was delivered, or punishment (−1) if odour 2 was delivered in step 2, and end of trial was indicated. No lick always led to 0 reward. Overall, a correct response for a trial in either block is defined as a lick for cue 1 or a no lick for cue 2 in the final reward step, while requiring no-lick in all other time steps. Block switches occurred using the same transition rules as in the experiment, described above.

Basic RL agent (Tabular SARSA algorithm): Each of the 2 visual and 2 olfactory cues, as well as the end-of-trial cue, was considered a state. The possible actions were lick and no-lick. A Q-value $Q(s, a)$ table was constructed with entries for each combination of 4 states (leaving out end-of-trial) and 2 actions, denoted by $s$ and $a$, initialized to zeros. The Q-value $Q(s, a)$ represented the expected total reward till the end of the trial, given the cue $s$ and taking action $a$ at the current step. The Q-value for the end of trial cue was set to 0. As each cue $s$ was encountered, the agent responded with an action $a$ according to an $\epsilon$-greedy policy. i.e., a random action was taken with probability $\epsilon$, otherwise the action that yielded the maximum Q-value for the current cue $s$ was taken. The Q-table was updated as per the SARSA (State-Action-Reward-State-Action) algorithm using the temporal difference (TD) error

$$\delta \equiv r + Q(s', a') - Q(s, a) \tag{1}$$

multiplied by a learning rate $\alpha$

$$Q(s, a) \leftarrow Q(s, a) + \alpha\delta \tag{2}$$

Cues $s', s$ and actions $a', a$ correspond to the current, previous time steps respectively.

Belief State RL agent: Two Q-value tables were constructed and initialized to zero, corresponding to the visual and olfactory blocks, each of size 4 states by 2 actions. The agent also had a belief $b$ about being in a visual block $v$ versus an olfactory block $o$, which was represented as a discrete probability distribution $b \equiv (p(v), p(o))$. At every step of the trial, the agent assumed that the current block was

either visual or olfactory depending on which probability $p(v)$ or $p(o)$ was higher, and took an action according to an $\epsilon$-greedy policy based on the Q-table corresponding to the assumed block. This Q-table was updated similar to the basic agent using the TD-error $\delta$ multiplied by a learning rate $\alpha$.

At the end of every trial, a block mismatch signal $\chi \equiv d - b$ was computed as the difference between the detected block $d$ (represented as $(1,0)$ or $(0,1)$ for visual or olfactory block respectively depending on whether the cue just before the end of the trial was a visual or odour cue), and the agent's belief $b$. A noise-corrupted version of this block mismatch signal $\chi' = \chi(1 + \beta\xi)$ was computed, where $\xi$ was a Gaussian random variable with mean 0 and variance 1, and $\beta$ was a noise factor parameter. The agent's belief was updated as:

$$b' = b + \zeta\chi' \tag{3}$$

where $\chi'$ was multiplied by a belief switching rate $\zeta$, and then each component of $b'$ was clipped to be greater than 0 and clipped $b'$ was normalized to yield a probability distribution $b$ which served as the belief for the next trial. After training, updating of Q-value tables was turned off, and only belief updates were carried out.

Model fitting: Each model was simulated with learning via SARSA for $N/2$ time steps. After this training, we obtained simulated behaviour data for a further $N/2$ time steps, for fitting to experimental data of trained mice. For Basic RL model fitting, exploration and updation of Q-value tables via SARSA parametrized by $\epsilon$ and $\alpha$ were kept the same as during training. For Belief-state RL model fitting, Q-values were no longer updated i.e., $\alpha = 0$ after training, only the belief state was updated, and exploration was kept on. We confirmed that keeping Q-value updating on after learning, at $\alpha = 0.1$ as during training, did not have a noticeable effect on the results.

We minimized the root mean squared error (RMSE) between the experimental and simulated $p(lick|cue)$. For all fits, we performed a global grid search within reasonable parameter ranges, followed by a local minimization starting from the best parameter sets obtained from the grid search. The fits were performed using 5-fold cross validation, where for each fold, we fit the parameters of the model to 4/5th of the data and tested on the held out 1/5th of the data. RMSE mean for each fold i.e., each training and test split was calculated across 5 seeds (2 with the first variation + 3 seeds with the second variation in a similar ratio as the number of experimental sessions on the two variations of the block transitions described above). To select between the models, the RMSE mean ± SD across these 5 folds, computed on the above RMSE mean across seeds, were reported and compared as below.

For the default basic RL model, we fit 2 parameters: exploration rate $\epsilon$ and the learning rate $\alpha$. This model was unable to reproduce the rapid block transitions observed in the data, since both the exploration rate and the learning rate needed to be very high to rapidly explore actions and learn a different reward structure after a transition, but a high exploration rate was inconsistent with the steady-state experimental data. For the default belief-state RL model, we fit 4 parameters: belief switching rate $\zeta$, the noise factor $\beta$ for the prediction-error signal, exploration rate $\epsilon$, and a different exploration rate $\epsilon'$ to account for enhanced licking to visual cue 2 in the olfactory block. The learning rate $\alpha$ was fixed at 0.1 during training, but was set to zero for the fitted data, as the belief switching rate played a much stronger role in rapid switching between blocks. The RMSE mean ± SD on training and test splits were $0.157438 \pm 0.004906$ and $0.193520 \pm 0.010869$ for the default basic RL agent with 2 parameters, and $0.087012 \pm 0.005060$ and $0.137714 \pm 0.014658$ for the default belief-state RL with 4 parameters, leading us to choose the Belief-state RL model over the Basic one.

We also fitted a Basic RL model with 3 parameters: exploration rate $\epsilon$, learning rate $\alpha$, and independent exploration rate $\epsilon'$ on receiving visual cue 2, which yielded RMSE mean ± SD on 5-fold training and test

splits as $0.161927 \pm 0.006923$ and $0.197955 \pm 0.011329$. Further, we fitted a Belief-state RL model with 3 parameters: belief switching rate $\zeta$, noise factor $\beta$, and a common exploration rate $\epsilon$, which yielded RMSEs as $0.095403 \pm 0.006537$ and $0.143631 \pm 0.015731$. This shows that $\epsilon'$, the enhanced exploration rate to visual cue 2, does not play a major role in selecting between Basic vs Belief-state RL models.

Simulated $p(lick|cue)$ using the best fit parameters (on the full dataset) for each model for one seed are shown in Fig. 1G and Supplementary Fig. 3C middle and bottom. The prediction-error signal for one-shot versus slower switches made by the agent, shown in Fig. 6C, was computed as the sum of the absolute values of the two components of the prediction-error signal $\chi$ at the end of the first trial following the block transition. For each of both types of transitions, only 70 transitions were chosen randomly from the simulated data, similar to the number of transitions in the experimental data, for plotting and significance testing.

For fitting the behaviour during ACC silencing, which was a separate experimental dataset, first we fitted the default 4 parameters for the belief-state RL model on the behaviour data without ACC silencing. Then keeping these parameters fixed, we fitted the behaviour data with ACC silenced (Supplementary Figs. 2d and 3f), using two parameters – factors on the prediction-error signal for odour and visual trials. These factors signified how much the prediction-error signal reduced on silencing the ACC.

Fitted parameters are shown in Table 1. Since these fits were not for model selection, all of the data was fit, minimizing mean RMSE across 5 seeds (both variations of the task included as described above).

## Imaging data analysis

Image stacks were corrected for motion, and regions of interest (ROIs) were selected for each active neuron in each session using Suite2p[76]. Each site yielded between 129 and 925 neurons, median = 499 neurons. Raw fluorescence time series F(t) were obtained for each neuron by averaging across pixels within each ROI. Baseline fluorescence F0(t) was computed by smoothing F(t) (causal moving average of 0.75 s) and determining for each time point the minimum value in the preceding 60 s time window. The change in fluorescence relative to baseline, ΔF/F, was computed by taking the difference between F and F0, and dividing by F0.

To identify prediction-error neurons, we selected neurons which responded significantly differently to the odour prediction-error event when compared to both the expected arrival of the odour and when no odour was expected (Fig. 2B). We defined three epochs each lasting 1.5 s and measured the average neural activity in these epochs: 1) Odour prediction-error trials, starting 2.0 s after the offset of the first visual stimulus following a switch from an odour block to a visual block, provided the mouse did not already lick to the preceding visual stimulus (2.0 s was the average delay in the imaging sessions from the visual stimulus offset to the odour stimuli onset). 2) Stable odour block trials from the end of the preceding odour block (when an odour is expected and received) aligned to the onset of the odour stimuli, following a correctly ignored visual grating. 3) No odour expected trials, when no odour is expected following a visual stimulus, 2.0 s following the offset of an unrewarded visual stimulus. Trials were taken from subsequent visual block up to 10 trials before the end of the block. In epochs 2 and 3, we averaged up to 7 trials of each condition for each block transition (median 7 trials). We compared the neural activity in different epochs with a Wilcoxon rank-sum test with the number of samples equal to the number of block transitions. Prediction-error neurons were defined as neurons with significantly different activity in odour prediction-error trials, when compared to both of the other conditions. We repeated the analysis without averaging multiple trials in epochs 2 and 3 and still obtained a significantly larger fraction of prediction-error neurons in the ACC compared to V1 (Chi-squared test

**Table 1 | Best-fit parameters for each model**

| Type of model / experimental data | Fixed parameters | Fitted parameters | Mean ± SD of RMSEs across 5 seeds |
|---|---|---|---|
| Basic RL on primary dataset | $N = 2,000,000$ (1,000,000 during fitting) | $\epsilon, \alpha = 0.236, 0.9$ | $0.158488 \pm 0.018537$ |
| Belief-state RL on primary dataset | $N = 500,000$ $\alpha = 0.1$ during training, $= 0$ during testing | $\zeta, \beta, \epsilon, \epsilon' = 0.6858, 1.997, 0.049, 0.280$ | $0.080526 \pm 0.004806$ |
| Belief-state RL on ACC silencing dataset with ACC not silenced | $N = 500,000$ $\alpha = 0.1$ during training, $= 0$ during testing | $\zeta, \beta, \epsilon, \epsilon' = 0.7698, 1.987, 0.1072, 0.4516$ | $0.053635 \pm 0.001487$ |
| Belief-state RL on ACC silencing dataset with ACC silenced | $N = 500,000$ $\alpha = 0.1$ during training, $= 0$ during testing $\zeta, \beta, \epsilon, \epsilon' = 0.7698, 1.987, 0.1072, 0.4516$ | ACC inhibition factors on prediction error for visual and odour trials = 0.2143, 0.5016 | $0.0742734 \pm 0.000907$ |

of proportions $P < 0.0001$). Similar criteria were used for the visual to odour block transitions. Positively and negatively responsive prediction-error neurons were those in which the response to the odour prediction error was largest or smallest of the three conditions respectively. Two other combinations were observed, first with the odour condition significantly higher and no odour condition significantly lower compared to the prediction-error condition (99 neurons), and second, the reverse of this (163 neurons). To test whether activity in prediction-error neurons could predict subsequent switching, average activity between 0–1.5 s in odour prediction-error trials was compared between one-shot switches (in which the mouse responded correctly to the subsequent visual grating) and slower switches (in which the mouse continued to miss at least the next visual grating) using a Wilcoxon signed-rank test.

Decoding analysis (Figs. 4D, 7G, I) was performed by training a binary logistic regression classifier to decode the stimulus or block identity from the vector of ΔF/F values at each frame. Decoding performance was assessed using stratified 5-fold cross validation and taking the mean accuracy across the 5 test sets. Stimuli classes were evenly balanced by randomly subsampling the larger class. We applied an L2 regularisation penalty to reduce overfitting.

For identification of striatal projecting neurons, a brief dual channel recording of the imaging planes was acquired before each imaging session at an excitation wavelength of 830 nm. Following segmentation, imaged neurons co-expressing Alexa-647 were identified using this recording, and confirmed using a detailed anesthetised dual channel z-stack taken at the end of all imaging sessions. To calculate confidence intervals for the percentage of prediction-error neurons in the non-striatal projecting ROIs (Fig. 4M) a percentile bootstrap method was used, resampling with replacement a number of ROIs equivalent to the size of the striatal projecting ROI population 10,000 times. The proportion of prediction-error neurons was then calculated and the $0.5^{th}$ and $99.5^{th}$ percentiles of this distribution of proportions was calculated to obtain the 99% confidence interval.

To assess the proportion of neural activity which was attributable to overt behaviour recorded during our task (Supplementary Fig. 5C, D, E) a linear model was fit using ridge regression to predict neural activity. The model was constructed by combining multiple sets of variables into a design matrix, to capture signal modulation by the following different task or behavioural events: 2 visual stimuli, 2 odour stimuli, reward delivery, licks, running speed, block type, and an interaction term for visual stimuli and block type. Each stimulus/event variable was structured to capture a time-varying event kernel. Variables therefore consisted of a vector of the relevant stimulus/event, and copies of this vector, each shifted in time by one frame for specific durations. For sensory stimuli, the time-shifted copies ranged up to 2 s after the original. For motor events (running and licking) the time-shifted copies spanned the frames from 0.5 s before until 2 s after the original. The model was fit with 5-fold cross validation and the coefficient of determination ($R^2$) was calculated based on the predictions of

the model on held out data not used during training. We then assessed the predictive power of the behavioural model variables by comparing the $R^2$ value for the full model to a model without the running and licking predictors.

## Optogenetic activation and inhibition of VIP interneurons during ACC imaging

To perturb VIP neurons expressing Chrimson, ArchT, or tdTomato concurrently with 2-photon imaging, a 639 nm laser (OBIS, Coherent) was used to deliver light via a 200 μm diameter, 0.39 NA optic fibre (Thorlabs) positioned around 3 mm from the posterior edge of the microprism, at a 30° angle relative to the coverslip. The laser and stimulus monitors were blanked during the linear phase of the resonant scanner to allow quasi-simultaneous two-photon imaging and optogenetic activation. The effective maximum output power from the optic fibre was 5.3 mW. During an optogenetic calibration session in the dark, 5 laser powers (20%, 40%, 60%, 80%, and 100% of maximum) were pseudorandomly applied to the coverslip for 1.5 s, with a 5 s interval between each. During switching sessions, two laser epochs were used, with 5.3 mW power only. In the peri-stimulus epoch, the laser began 0.1 s before the visual stimulus and continued until the end of the visual stimulus. In the inter-trial-interval epoch, the laser began at the offset of each visual stimulus and continued until the onset of the next visual stimulus.

## Pupil tracking

Eye recordings were acquired using a monochrome USB2.0 video camera (The Imaging Source) with a 50 mm 2/3" format 5-megapixel lens (Computar), set to acquire at 320x240 (Y800) resolution and 30 frames per second. Frames were triggered using an Arduino Uno microcontroller board (Arduino) to ensure a constant acquisition rate. Pupil data were extracted using DeepLabCut[77] for 2D marker tracking, with markers set to track vertical and horizontal boundaries of the eye and pupil throughout the recording. Frames that coincided with blinks were removed based on changed in vertical size of the eye, and pupil width was calculated from the remaining frames.

## Optogenetic silencing of ACC activity

Once mice had learned the full switching task, optogenetic silencing of ACC neurons was performed by connecting the optic fibre cannulae to a blue LED (470 nm, Thorlabs), and delivering light while the mouse performed the task. Before implantation, each optic fibre was confirmed to have an effective power output of >1 mW after cutting. Light was delivered either throughout the session (pulsed at 40 Hz[78]), 0.5 s before each visual stimulus continuing to the end of the stimulus ('peri-stimulus'), or from the end of each visual stimulus until the beginning of the next visual stimulus ('inter-trial-interval', ITI). These three epochs were used to silence both ACC and PL on different days, creating 6 silencing conditions. These conditions were pseudorandomly chosen across consecutive days, with the order different

between mice but the time between repeated conditions kept constant, and with control no-light sessions interspersed every third session. For one-trial un-silencing, the light was continuously pulsed throughout the session as above, but this was paused at the end of the last trial in the odour block, and resumed at the start of the second trial of the visual block. In the peri-stimulus and ITI epochs, light power was ramped down for the final 200 ms of each pulse. For ACC silencing during learning, the silencing group included all 8 optogenetic mice, and the non-silenced controls were the 10 imaging mice. Light-only controls were performed in PV-Cre mice not expressing channelrhodopsin ($N$ = 3 mice). No differences were found in these mice in the number of trials to transition between blocks with and without the light stimulation ($P$ > 0.05, Wilcoxon rank sum test).

### Pharmacological silencing of ACC activity

For the muscimol silencing experiments, 4 male wildtype C57Bl/6j mice were implanted with bilateral infusion cannulae (−1.2 mm DV, +0.7 mm AP, ± 0.5 mm ML, bilaterally). After training in the switching task, mice were infused bilaterally with 300 nl muscimol (Sigma, 1 μg/μl) or saline at a rate of 0.25 μl/min using a 1 μl syringe (Hamilton) and syringe pump (World Precision Instruments SP100IZ) 30 min before the start of a switching session. We waited 5 min after the syringe pump had finished to allow full infusion of the drug before disconnecting the cannulae.

### Silencing data analysis

Behavioural d-primes in each silencing condition were calculated by taking performance in stable periods of blocks, outside of transition periods[74,75]. Switching speeds were defined as the number of trials that elapse before a mouse correctly responded to three rewarded visual gratings in a row, either licking to the grating after a switch to a visual block or by ignoring the grating after a switch to an odour block. A 'fluke' switch was defined as a switch in which the mouse correctly licked to the very first rewarded grating in a new visual block, before any evidence of the switch had been received. These were interpreted as exploratory licks, visible in the histograms in Fig. 1F and Supplementary Fig. 3E at trial 0, and all such transitions were excluded from analysis of switching speeds.

### Reporting summary

Further information on research design is available in the Nature Portfolio Reporting Summary linked to this article.

## Data availability

The data that support the findings of this study are available at https://figshare.com/projects/Prediction-error_signals_in_anterior_cingulate_cortex_drive_task-switching/211438. Source data are provided with this paper.

## Code availability

The code generated in this study is available at https://github.com/adityagilra/BeliefStateRL with DOI release https://doi.org/10.5281/zenodo.12636612.

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

## Acknowledgements

We thank John Duncan, Apoorva Bhandari, Dimitar Kostadinov and Sonja Hofer for discussions and comments on the manuscript, and Thomas Mrsic-Flogel, Petr Znamenskiy, Anil Seth, Jasper Poort and Juan Burrone for discussions of the results in this manuscript. We thank Marian Fernandez-Otero for technical assistance and Robert Taylor for assistance in pilot experiments. This work was supported by the Wellcome Trust (AGK, 206222/Z/17/Z), the BBSRC (AGK BB/S015809/1), start-up funds from the CDN, King's College London (AGK), and the MRC CNDD PhD programme (MH).

## Author contributions

N.C. and A.G.K. designed the experiments. N.C. performed the experiments and analysed the data. M.H. performed the widefield imaging and contributed to imaging data analysis. D.M.-J. performed the V1 recordings and contributed to imaging and behavioural data analysis. AG built and analysed the RL model. A.G.K. and N.C. wrote the paper, with contributions from A.G., M.H. and D.M.-J. All authors contributed to discussions and commented on the manuscript.

## Competing interests

The authors declare no competing interests.
