## [Peer Review File · Nature Communications]

Prediction-error signals in anterior cingulate cortex drive task-switchingEditorial Note: This manuscript has been previously reviewed at another journal that is not operating a transparent peer review scheme. This document only contains reviewer comments and rebuttal letters for versions considered at *Nature Communications*.

REVIEWERS' COMMENTS

Reviewer #1 (Remarks to the Author):

The authors have revised their manuscript in the light of the previous round of comments. They have clarified the manuscript at several points.

In addition, the authors have added a significant new set of data examining the effect of activation or inhibition of inhibitory VIP cells in anterior cingulate cortex (ACC) on the activity found elsewhere in the ACC. Coles and colleagues report interesting decrements in prediction error signals thereby implicating the VIP neurons in the generation of the prediction error signals. This new addition to the report is the result of a carefully conducted set of additional experiments in which VIP cell manipulation is carried out unilaterally. This is a clever way to conduct the experiment because it means that behaviour itself does not change because there is still an intact ACC in one hemisphere. Therefore, the authors can be certain that the VIP cell manipulation is directly affecting prediction error signals in the activated/inhibited hemisphere. By contrast, had a bilateral manipulation been made that caused a behavioural change, then any change in neural activity could have been argued to be the secondary consequence of the behavioural change rather than directly due to the VIP change itself.

In summary, the manuscript is strengthened and improved and should now be published in *Nature Communications*.

Reviewer #2 (Remarks to the Author):

The authors did a reasonable job at pointing out various existing and new analyses that support their viewpoint that the signal reflects a prediction error. Due to the task design and some of the neural responses shown (e.g. the comment regarding the absence of an odor should still be represented even if the mouse does not switch its behavior), I still have doubts regarding the interpretation. But at this point I do not think this can be resolved by further discussions. The behavioral and neural data themselves are of decent quality, so the readers can look and judge for themselves.

I should also note the new data added to photoactivate or photoinhibit VIP interneurons is interesting and add to the overall story and strengthen the novelty.